# Targeted delivery of celastrol to mesangial cells is effective against mesangioproliferative glomerulonephritis

Ling Guo[1], Shi Luo[1], Zhengwu Du[1], Meiling Zhou[1], Peiwen Li[1], Yao Fu[1], Xun Sun[1], Yuan Huang[1] & Zhirong Zhang[1]

Mesangial cells-mediated glomerulonephritis is a frequent cause of end-stage renal disease. Here, we show that celastrol is effective in treating both reversible and irreversible mesangioproliferative glomerulonephritis in rat models, but find that its off-target distributions cause severe systemic toxicity. We thus target celastrol to mesangial cells using albumin nanoparticles. Celastrol-albumin nanoparticles crosses fenestrated endothelium and accumulates in mesangial cells, alleviating proteinuria, inflammation, glomerular hypercellularity, and excessive extracellular matrix deposition in rat anti-Thy1.1 nephritis models. Celastrol-albumin nanoparticles presents lower drug accumulation than free celastrol in off-target organs and tissues, thereby minimizing celastrol-related systemic toxicity. Celastrol-albumin nanoparticles thus represents a promising treatment option for mesangioproliferative glomerulonephritis and similar glomerular diseases.

[1] Key Laboratory of Drug Targeting and Drug Delivery Systems, Ministry of Education, West China School of Pharmacy, Sichuan University, Chengdu 610041, China. Correspondence and requests for materials should be addressed to Z.Z. (email: zrzzl@vip.sina.com)

Glomerulonephritis (GN) refers to a category of immunologically mediated glomerular injuries characterized by infiltration of circulating inflammatory cells, proliferation of glomerular cells and accumulation of extracellular matrix (ECM)[1], which often leads to glomerulosclerosis and end-stage renal disease[2]. According to the statistics by the US Centers for Disease Control and Prevention, GN and related kidney diseases were the 9th leading cause of death in the US in 2013[3]. Pharmacological treatments against inflammation and glomerular disorders may slow GN progression and related mortality.

Natural products constitute a great source for seeking potential therapeutic candidates. The traditional Chinese medicine, Thunder of God Vine (TGV) and its formulations, have long been used to treat GN in China[4–8]. Celastrol (CLT), a pentacyclic triterpene extracted from TGV, is a potent immunosuppressive, anti-inflammatory and anticancer agent[9]. Due to the abundance of CLT in TGV formulations[10, 11], we hypothesized that CLT might be the biologically active component in the treatment of GN.

To prove this hypothesis, we examined the therapeutic effects of CLT in a reversible and an irreversible rat model of anti-Thy1.1 nephritis, which are well-established animal models for mesangioproliferative glomerulonephritis (MsPGN)[12]. Mycophenolic acid (MPA), as a beneficial agent against anti-Thy1.1 nephritis[13, 14], was selected as the standard treatment control. We obtained encouraging results that CLT significantly attenuated

**Fig. 1** Early CLT treatment shows dose-dependent efficacy in the reversible model. **a** Effects of MPA (30 mg kg⁻¹) and CLT (LD-CLT, 1 mg kg⁻¹; MD-CLT, 2 mg kg⁻¹; HD-CLT, 3 mg kg⁻¹) on 24-h urinary protein excretion in anti-Thy1.1 nephritic rats on day 5 after disease induction. **b** Glomerular histology revealed by PAS staining of kidney tissue sections from anti-Thy1.1 nephritic rats on day 5 after early treatment with MPA or different doses of CLT. *Scale bars*, 20 µm. **c** Effects of MPA and different doses of CLT on total glomerular cellularity on day 5 after disease induction. For each animal group, 150 glomeruli were selected and total glomerular cells were counted using cellSens Standard digital imaging software (Olympus). **d** Effects of MPA and different doses of CLT on ECM accumulation on day 5 after disease induction. For each animal group, 150 glomeruli were analyzed and ECM deposition was graded semiquantitatively as described in Methods. In panels **a**, **c**, and **d** data are mean ± s.d. (n = 5), results are representative of two independent experiments. *P < 0.05 vs. PBS group; #P < 0.05 vs. MPA group. Statistical significance was determined by one-way ANOVA with Tukey post hoc test. **e** Flow diagram of the early treatment of CLT or MPA against the reversible anti-Thy1.1 nephritis. *Black triangle* denotes intravenous treatment of CLT or MPA; *black star* denotes time points of nephrectomy while respective animals were sacrificed. A detailed description is given in Methods

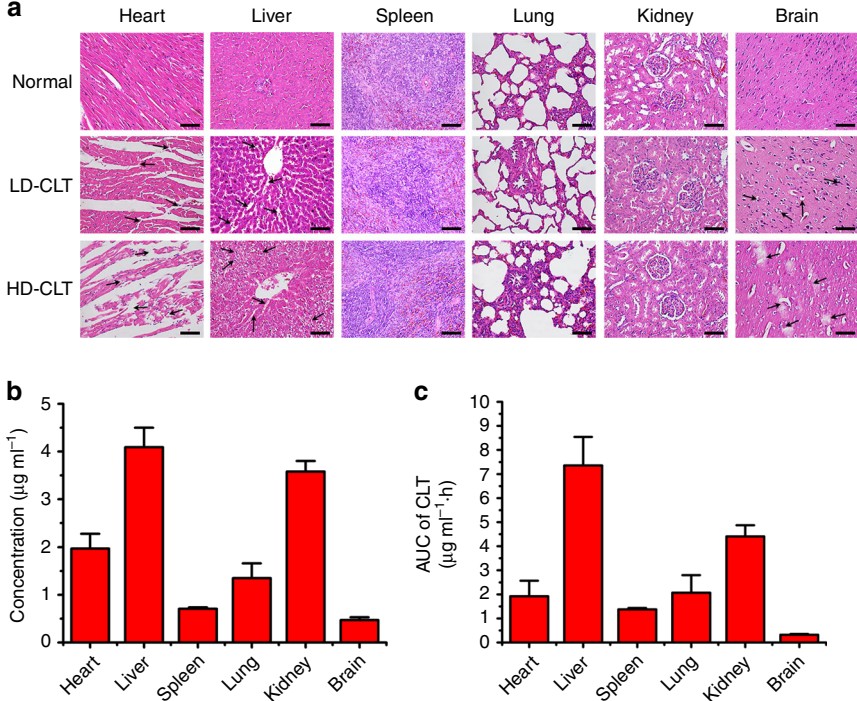

**Fig. 2** CLT induces severe cardiotoxicity, hepatotoxicity, or neurotoxicity. **a** Hematoxylin & eosin (H&E) staining assay of heart, liver, spleen, lung, kidney, and brain on day 5 post-exposure to LD-CLT (1 mg kg$^{-1}$) or HD-CLT (3 mg kg$^{-1}$). *Black arrows* in heart samples indicate atrophy of myocardial cells and myofibrillar loss; *arrows* in liver samples of LD-CLT-treated group indicate atrophy of hepatic cells and dilatation of blood sinus, and *arrows* in HD-CLT-treated group indicate the edema and atrophy of hepatic cells; *arrows* in brain samples of LD-CLT-treated group indicate pyknosis of neuron and red neurons, and *arrows* in HD-CLT-treated group indicate liquefactive necrosis foci. *Scale bars*, 50 μm. **b** Tissue distribution in rats at 5 min after intravenous administration of CLT. **c** AUC$_{0-4h}$ of CLT in different organs. In panels **b**, **c** data are mean ± s.d. ($n = 5$), results are representative of two independent experiments

proteinuria, inflammation, glomerular hypercellularity, and ECM deposition in anti-Thy1.1 nephritis (Fig. 1; Supplementary Figs. 1, 2, 4–9), indicating that CLT was a main contributory ingredient involved in TGV formulations in the treatment of MsPGN. Specifically, 3 mg kg$^{-1}$ CLT was proven much more effective than 30 mg kg$^{-1}$ MPA, suggesting that CLT as a single compound might be a promising candidate for MsPGN therapy. However, CLT was reported to induce severe cardiotoxicity in zebrafish embryo at micromolar concentrations[15]. Also, the intraperitoneal injection of free CLT at the dose of 1 mg kg$^{-1}$ led to severe lymphocyte infiltration in liver sinuses in mice[16]. Therefore, we aimed to develop a targeted approach that can deliver CLT preferentially to the disease site, reducing the risk of systemic toxicity.

Glomerular mesangial cells may be potential cellular targets for treating MsPGN because their malfunctions result in the initiation and progression of MsPGN[17]. Selectively delivering CLT to mesangial cells might help alleviate local mesangial cell responses, while minimizing off-target drug exposure and reducing systemic toxicity. Nanoparticles appear a vehicle of choice for targeted drug delivery owing to their size-dependent accumulations in organs such as liver and lung[18, 19]. Gold nanoparticles with a defined size of ~75 ± 25 nm were shown to specifically accumulate in mesangial cells in mice[20]. However, whether a nanoscale system can selectively deliver therapeutics to mesangial cells remains to be explored.

In the present study, we select human serum albumin (HSA) to produce albumin nanoparticles (ANs) with defined sizes to deliver CLT selectively to mesangial cells. To screen the optimal particle size to achieve mesangial cells targeting, we first study the impact of nanoparticle size on ANs localization at mesangial cells.

Then, we produce CLT-loaded albumin nanoparticles (CLT-AN) with a well-defined size, and elucidate its targetability to mesangial cells, therapeutic efficacy, and toxicity. We also investigate the possible therapeutic mechanisms in anti-Thy1.1 nephritic rats and compare biodistribution behaviors between CLT-AN and free CLT. CLT-AN showing excellent mesangial cells-targetability attenuates glomerular lesions in rat anti-Thy1.1 nephritis models via anti-inflammatory, anti-proliferative, and anti-fibrotic mechanisms. Also, CLT-AN presents lower drug concentration than free CLT in off-target tissues, thus reducing CLT-related systemic toxicity. To our knowledge, this is the first study on the therapeutic effect of CLT against MsPGN and the first report of ANs for mesangial cells-targeted drug delivery to improve the efficacy and safety of CLT.

## Results

**CLT attenuates glomerular lesions in anti-Thy1.1 nephritis.** In the reversible rat model of anti-Thy1.1 nephritis, the early treatment starting from day 0 showed that LD-CLT (low dose of CLT, 1 mg kg$^{-1}$) attenuated proteinuria to a similar extent as MPA at 30 mg kg$^{-1}$, while HD-CLT (high dose of CLT, 3 mg kg$^{-1}$) was much more effective than either LD-CLT or 30 mg kg$^{-1}$ MPA (Fig. 1a). From periodic acid-Schiff reagent (PAS) stained sections, nephritic rats displayed prominent mesangial hypercellularity and ECM expansion on day 5 after disease induction, which were largely attenuated by MPA and CLT treatment (Fig. 1b). Semiquantitative analysis indicated fewer total cells per glomerular cross-section and lower mesangial matrix score in nephritic rats treated with MPA or CLT than in control animals treated with phosphate-buffered saline (PBS). Moreover, HD-CLT treatment showed the lowest values with

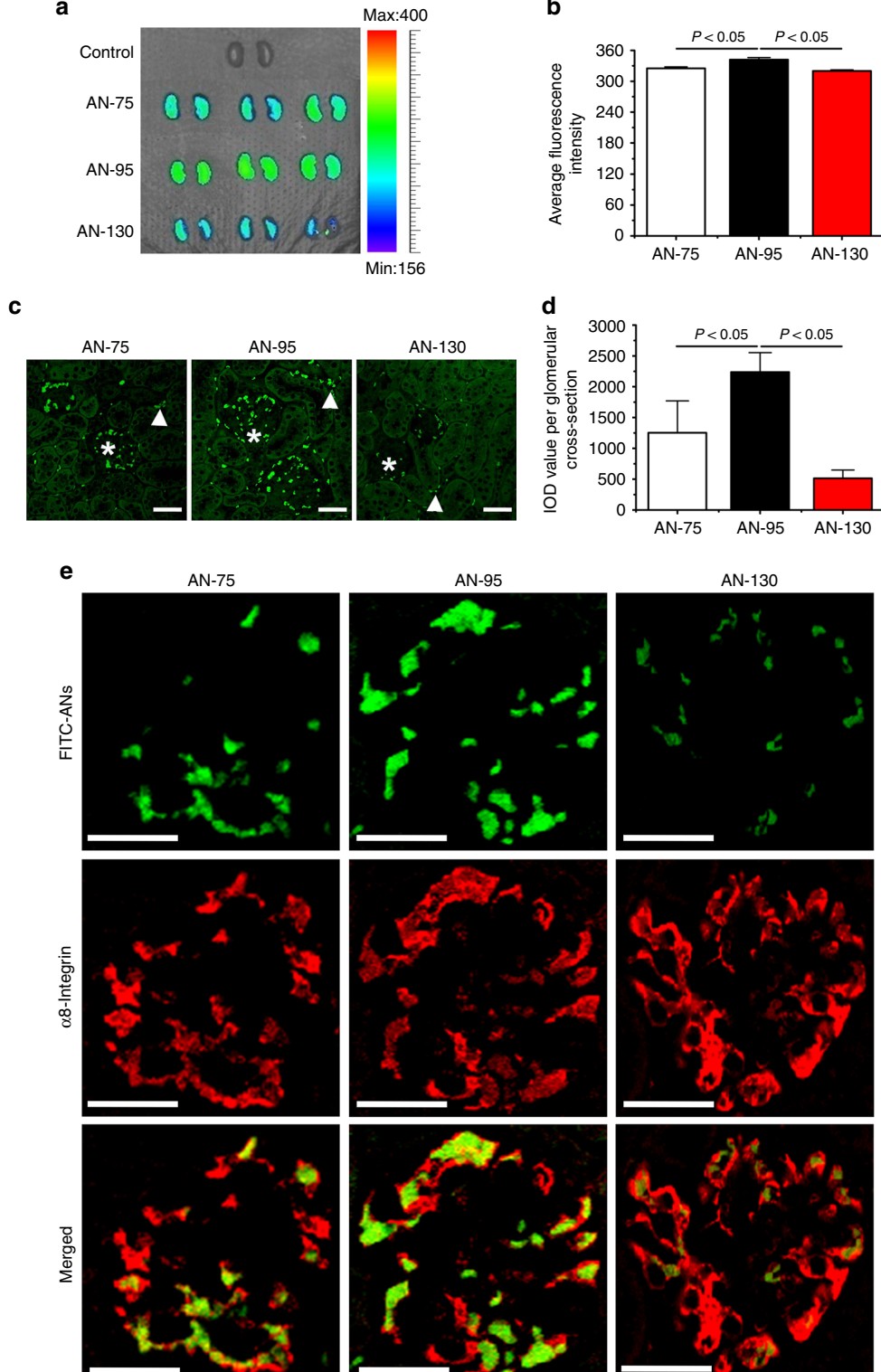

**Fig. 3** FITC-ANs size-dependently localize at the glomerular mesangial cells. **a** In vivo fluorescence images of kidneys excised at 5 min after tail vein injection of FITC-ANs. **b** Semiquantitative fluorescence intensity of excised kidneys in **a**. **c** Representative confocal images of kidney tissue sections taken at 5 min after tail vein injection of FITC-ANs. * and Δ mark glomeruli and peritubular interstitial space, respectively. *Scale bars*, 30 μm. **d** Integrated optical density (*IOD*) per glomerular cross-section representing the fluorescence intensity and spread of FITC-ANs in glomerulus as analyzed by cellSens Standard digital imaging software (Olympus). For each treatment group AN-75 to AN-130, 150 glomeruli were analyzed. In panels **b**, **d** data are mean ± s.d. (*n* = 3), results are representative of two independent experiments. Statistical significance was determined by one-way ANOVA with Tukey post hoc test.
**e** Representative confocal images of FITC-ANs localization at glomerular mesangial cells. Kidney tissue sections obtained at 5 min after tail vein injection of FITC-ANs (*green*) were counter-stained with NL637 conjugated antibody against the mesangial marker α8-integrin (*red*). *Scale bars*, 30 μm

35.3% and 65.4% reductions in total glomerular cell numbers and mesangial matrix score, respectively (Fig. 1c, d).

ED-1 has been used widely as a marker for rat macrophages[21]. PCNA, a proliferative cell nuclear antigen, is commonly used as a cell proliferation marker[22]. The α-smooth muscle action (α-SMA) expressed in mesangial region is associated with the mesangial cell proliferation and represents a marker of cell activation[23]. Per immunostaining results, induction of anti-Thy1.1 nephritis resulted in obvious infiltration of monocytes/macrophages and upregulated levels of α-SMA and PCNA in glomeruli, while MPA and CLT markedly downregulated these markers (Supplementary Fig. 1a). Nephritic rats treated with MPA or CLT showed fewer ED-1-positive cells per glomerular cross-section on day 1 after disease induction, and lower α-SMA score and fewer PCNA-positive cells per glomerular cross-section on day 5, moreover, HD-CLT treatment had the least values among all the treatment groups (Supplementary Fig. 1b–d).

Type I collagen (Col I), type IV collagen (Col IV), and fibronectin-1 (FN-1) secreted by mesangial cells constitute ECM proteins deposited within glomerular mesangium[24]. Per immunostaining results, all three proteins were overproduced in glomeruli on day 5 after disease induction, which were significantly reduced following treatment with MPA or CLT (Supplementary Fig. 2a). Semiquantitative analysis demonstrated lower ECM protein scores in nephritic rats treated with MPA or CLT compared to PBS control, and HD-CLT treatment showed the least scores among all treatment groups (Supplementary Fig. 2b–d).

Marked proteinuria, glomerular infiltration of circulating monocytes, glomerular hypercellularity and ECM accumulation were observed in anti-Thy1.1 nephritic rats at day 2 after disease induction compared to normal control (Supplementary Fig. 3). Also, delayed CLT treatment initiating at day 2 after disease induction significantly attenuated proteinuria and glomerular lesions in anti-Thy1.1 nephritis rats (Supplementary Figs. 4, 5, and 6), showing consistent trends compared to the early treatment.

Taken together, we first demonstrate the excellent therapeutic efficacy of both early and delayed CLT treatment against proteinuria and acute glomerular lesions in reversible anti-Thy1.1 nephritis model. However, the proteinuria, and glomerular architecture in the present model return to normal within 2 to 4 weeks[12], which is not appropriate for evaluating the effects of CLT on progressive glomerular lesions in MsPGN. Next, a single dose of OX7 was administered to unilaterally nephrectomized rats to induce an irreversible model of anti-Thy 1.1 nephritis[12]. We further examined the efficacy of CLT against the progression of irreversible mesangial lesions. On day 14 after disease induction, nephritic rats treated with PBS displayed marked proteinuria, infiltration of monocytes/macrophages, mesangial cell proliferation, and ECM expansion, which were significantly suppressed by CLT treatment (Supplementary Figs. 7, 8, and 9). Moreover, HD-CLT showed the least glomerular lesions among all the treatment groups.

Overall, our data support the beneficial effects of CLT to suppress proteinuria and attenuate acute and progressive glomerular pathologic insults in anti-Thy1.1 nephritis. CLT shows dose-dependent efficacy, which holds great promise in the treatment of MsPGN.

**CLT causes severe systemic toxicity.** Despite the excellent therapeutic efficacy of CLT, its potential toxicity remained largely unexplored. Here, we investigated the systemic toxicity of CLT at day 5 post-exposure to LD-CLT or HD-CLT (Fig. 2; Supplementary Fig. 10). As for renal function, neither LD-CLT nor HD-CLT significantly altered levels of blood urea nitrogen (BUN) or creatinine (CREA) (Supplementary Fig. 10a, b). Liver injury or dysfunction was evaluated by serum alanine transaminase (ALT), aspartate transaminase (AST), total bilirubin (TBiL) and international normalized ratio (INR)[25–27]. CLT showed dose-dependent increases in ALT, AST, and TBiL with no alterations in INR (Supplementary Fig. 10c–f). INR, standardization of prothrombin time measurement, has been used as an indicator to evaluate the ability of liver to synthesize blood coagulation factors and the severity of acute and chronic liver injury[25, 28]. Our results indicate CLT did not influence liver synthetic function although it may induce injury to the liver. Cardiotoxicity was evaluated by creatine kinase (CK) and lactate dehydrogenase (LDH), which are two common indexes for the diagnosis of cardiac disease[29, 30]. The levels of CK and LDH showed dose-dependent elevations in CLT-treated groups (Supplementary Fig. 10g, h), indicating the potential cardiotoxicity of CLT.

Per histological examinations, animals showed no visible lesions in the spleen, lung, or kidney after LD-CLT or HD-CLT treatment (Fig. 2a). However, CLT induced serious damages to heart, liver and brain. Atrophy of myocardial cells and myofibrillar loss were observed in animals treated with CLT, and increasing the dose aggravated the signs of cardiotoxicity (Fig. 2a). LD-CLT-treated group showed moderate atrophy of hepatic cells and dilatation of blood sinus, while HD-CLT-treated group displayed moderate edema and atrophy of hepatic cells (Fig. 2a). Besides, LD-CLT led to severe pyknosis of neuron and red neuron in the brain. Brain tissues exposed to HD-CLT even revealed the sign of liquefactive necrosis foci (Fig. 2a).

The biodistribution profiles of CLT in rats provide guidance for understanding its systemic toxicity. CLT administered via intravenous injection displayed extensive distributions in major organs (Fig. 2b, c), and the highest accumulation in liver may explain the serious hepatotoxicity of CLT. Accumulations of CLT were also observed in the heart or brain thus inducing cardiotoxicity or neurotoxicity. Although CLT accumulations to varying degrees were observed in the spleen, lung, and kidney, the CLT concentrations in these organs did not cause noticeable changes to tissue histology. Collectively, CLT led to severe cardiotoxicity, hepatotoxicity, or neurotoxicity mainly due to its off-target distributions.

**ANs size-dependently accumulate in mesangial cells.** To improve the safety and efficacy of CLT, we attempted to develop a mesangial cells-targeted drug delivery system based on sized ANs. Gold nanoparticles from 25 to 180 nm were shown to size-dependently target to the glomerular mesangium following intravenous administration, and 75-nm nanoparticles exhibited the maximal mesangial cellular deposition[20]. To examine whether ANs would accumulate at mesangial cells similarly as the rigid inorganic gold nanoparticles, we sought to elucidate the relationship between nanoparticle size and ANs localization at mesangial cells.

Fluorescein isothiocyanate (FITC)-labeled ANs (FITC-ANs) of various sizes were prepared: AN-75, ~ 75 nm; AN-95, ~ 95 nm; and AN-130, ~ 130 nm, and their detailed characterizations were provided in the Supplementary Fig. 11. Next, the ex vivo imaging of mice kidneys was performed at 5 min after intravenous injection of AN-75, AN-95, and AN-130. FITC-ANs extensively distributed throughout kidneys as compared to the control group, whereas AN-95 displayed significantly higher renal accumulation than the other two treatment groups (Fig. 3a). The semiquantitative analysis of mean fluorescence intensity also revealed that AN-95 achieved the most enhanced renal distribution among three treatment groups (Fig. 3b).

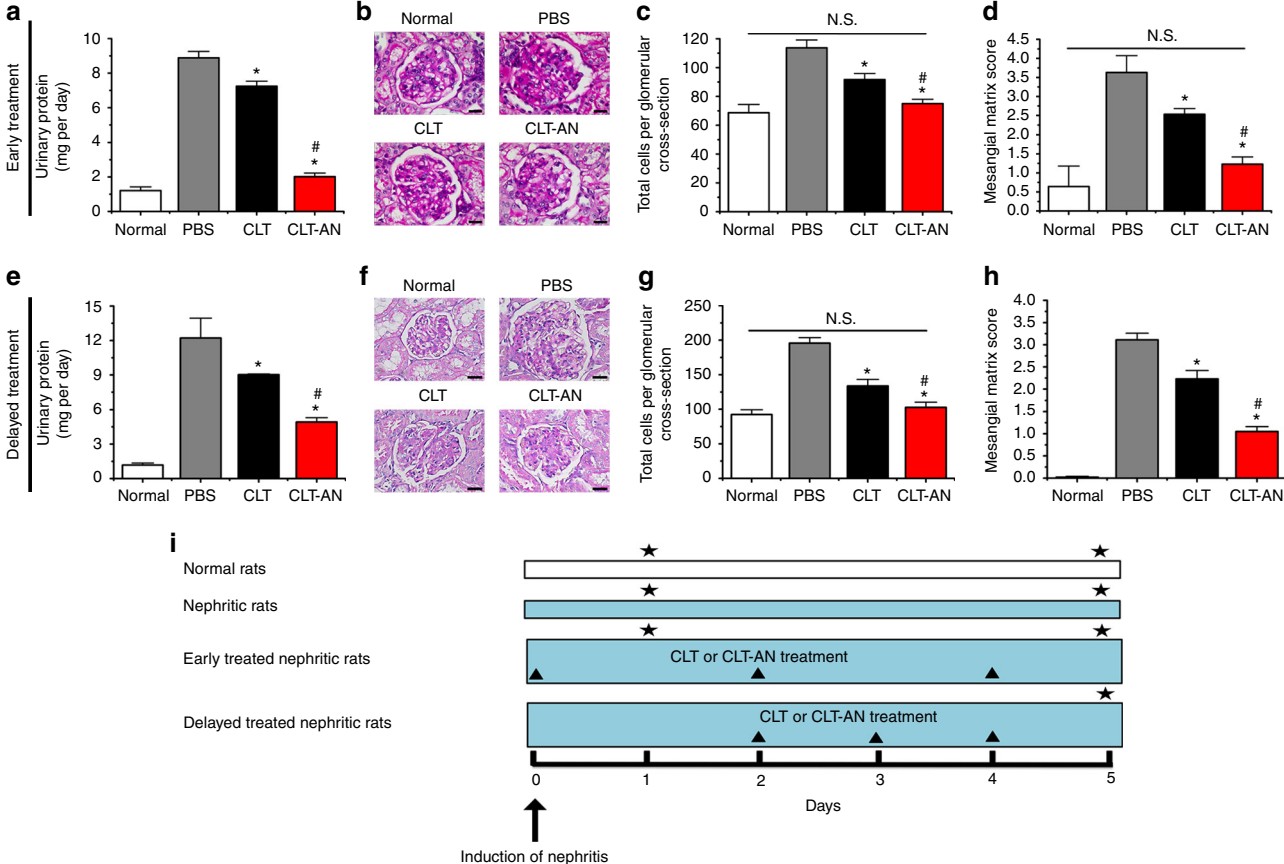

**Fig. 4** Enhanced efficacy of CLT-AN in the reversible model. **a** Effects of early CLT or CLT-AN treatment on 24-h urinary protein excretion in anti-Thy1.1 nephritic rats on day 5 after disease induction. **b** Glomerular histology revealed by PAS staining of kidney tissue sections from anti-Thy1.1 nephritic rats on day 5 after early treatment with CLT or CLT-AN. *Scale bars*, 20 μm. **c** Effects of early CLT or CLT-AN treatment on total glomerular cellularity on day 5 after disease induction. For each animal group, 150 glomeruli were selected and total glomerular cells were counted using cellSens Standard digital imaging software (Olympus). **d** Effects of early CLT or CLT-AN treatment on ECM deposition on day 5 after disease induction. For each animal group, 150 glomeruli were analyzed and ECM deposition was graded semiquantitatively as described in Methods. **e** Effects of delayed CLT or CLT-AN treatment on 24-h urinary protein excretion in anti-Thy1.1 nephritic rats on day 5 after disease induction. **f** Glomerular histology revealed by PAS staining of kidney tissue sections from anti-Thy1.1 nephritic rats on day 5 after delayed treatment with CLT or CLT-AN. *Scale bars*, 20μm. **g** Effects of delayed CLT or CLT-AN treatment on total glomerular cellularity on day 5 after disease induction. **h** Effects of delayed CLT or CLT-AN treatment on ECM deposition on day 5 after disease induction. In panels **a**, **c**, **d**, **e**, **g**, and **h** data are mean ± s.d. ($n = 5$), results are representative of two independent experiments. N.S., not significant; *$P < 0.05$ vs. PBS group; #$P < 0.05$ vs. CLT group. Statistical significance was determined by one-way ANOVA with Tukey post hoc test. **i** Flow diagram of the early and delayed treatments of CLT or CLT-AN against the reversible anti-Thy1.1 nephritis. *Black triangle* denotes intravenous treatment of CLT or CLT-AN; *black star* denotes time points of nephrectomy while respective animals were sacrificed. A detailed description is given in Methods

To understand the intrarenal accumulation features of FITC-ANs, we tracked the fluorescent signal present in kidney tissue sections from mice that received AN-75, AN-95, or AN-130. Consistent with previous findings[20], FITC-ANs mainly distributed within glomeruli and peritubular interstitial spaces, and no fluorescent signals except for an weak autofluorescence was detected in the tubular region (Fig. 3c). The integrated optical density (IOD) per glomerular cross-section representing the fluorescence intensity and spread of FITC-ANs in glomerulus indicated their size-dependent intraglomerular accumulation with AN-95 yielding the highest IOD value (Fig. 3c, d).

Further, immunofluorescence staining against α8-integrin, a mesangial marker[31], was used to examine the intraglomerular localization of FITC-ANs (Fig. 3e). FITC-positive areas of FITC-ANs in the glomeruli were mostly overlaid with α8-integrin staining, suggesting the mesangial cellular localization of FITC-ANs. Semiquantitative co-localization analysis revealed 33.7, 53.4, and 6.9% α8-integrin staining co-localization with the fluorescence of AN-75, AN-95, and AN-130, respectively. These results indicate organic ANs in the present study also

size-dependently accumulated at mesangial cells, and AN-95 achieved the highest accumulation.

The glomerular endothelium between glomerular capillaries and mesangium is fenestrated with pores ranging in diameters from 80 to 150 nm[32]. All three ANs should theoretically pass freely through the endothelial fenestrations and enter mesangium. However, AN-95 showed higher mesangial cellular localization than the other two formulations (Fig. 3). The cellular internalization of nanoparticles is related with routes of endocytosis, and particle size has been a main contributing factor in determining the internalization pathway[33]. To explore the potential mechanism of the size-dependent mesangial cellular retention of ANs, we investigated their internalization pathways in mesangial cells. The cellular uptake of AN-75, AN-95, or AN-130 was significantly inhibited at 4 °C (Supplementary Fig. 12a–c), indicating their endocytosis was energy-dependent. To elucidate the internalization pathways, different specific inhibitors were preincubated with HBZY-1 cells and their uptake rates were investigated by flow cytometry. Amiloride, nystatin, and chlorpromazine were employed to block macropinocytosis,

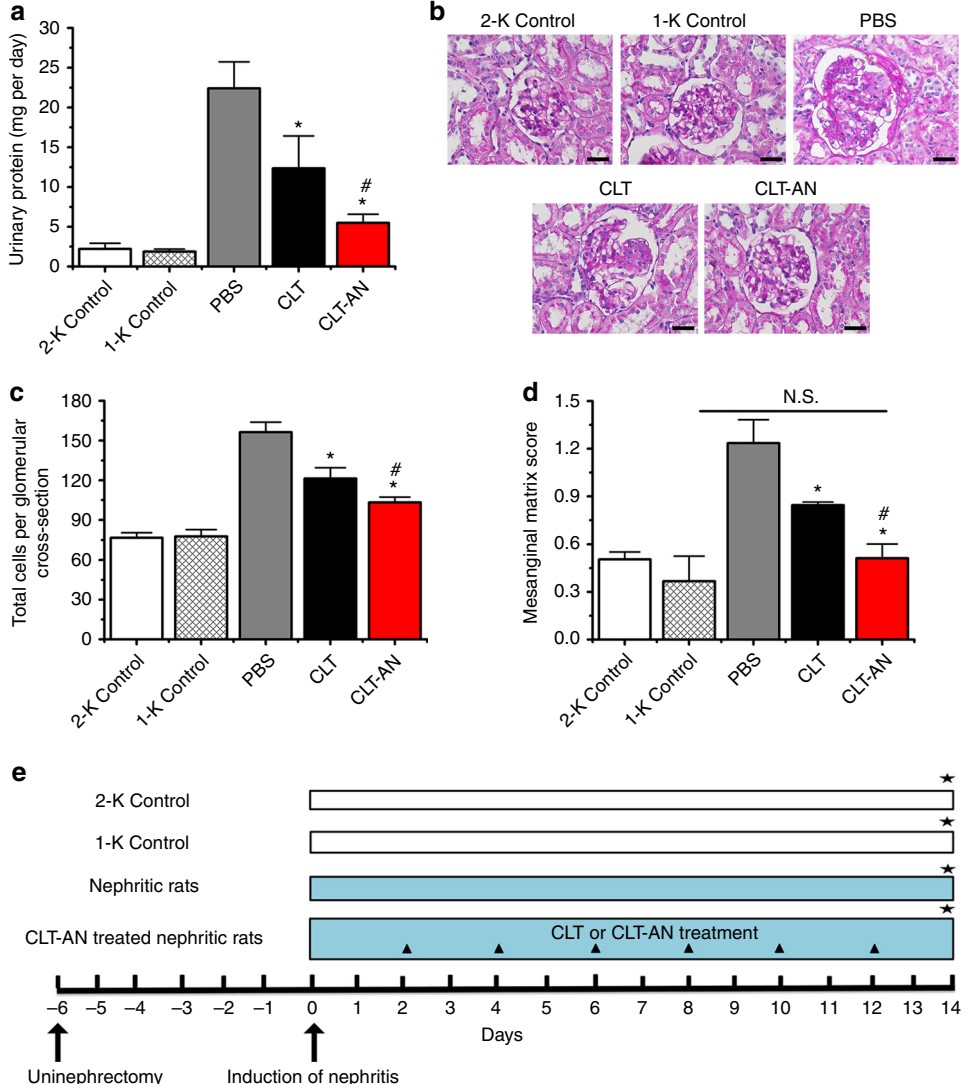

**Fig. 5** Enhanced efficacy of delayed CLT-AN treatment in the irreversible model. **a** Effects of delayed CLT or CLT-AN treatment on 24-h urinary protein excretion in anti-Thy1.1 nephritic rats on day 14 after disease induction. **b** Glomerular histology revealed by PAS staining of kidney tissue sections from anti-Thy1.1 nephritic rats on day 14 after delayed treatment with CLT or CLT-AN. *Scale bars*, 20μm. **c** Effects of delayed CLT or CLT-AN treatment on total glomerular cellularity on day 14 after disease induction. For each animal group, 150 glomeruli were selected and total glomerular cells were counted using cellSens Standard digital imaging software (Olympus). **d** Effects of delayed CLT or CLT-AN treatment on ECM accumulation on day 14 after disease induction. For each animal group, 150 glomeruli were analyzed and ECM deposition was graded semiquantitatively as described in Methods. In panels **a**, **c**, and **d** data are mean ± s.d. ($n = 5$), results are representative of two independent experiments. Nonnephrectomized two-kidney control (2-K Control) and uninephrectomized one-kidney control (1-K Control) served as controls. *$P < 0.05$ vs. PBS group; #$P < 0.05$ vs. CLT group. Statistical significance was determined by one-way ANOVA with Tukey post hoc test. **e** Flow diagram of the treatment of CLT or CLT-AN treatment against the irreversible anti-Thy1.1 nephritis. *Black triangle* denotes intravenous treatment of CLT or CLT-AN; *black star* denotes time points of nephrectomy while respective animals were sacrificed. A detailed description is given in Methods

caveolae- and clathrin-mediated endocytosis, respectively[34]. Compared with control, the uptake of AN-75 was significantly reduced by nystatin and amiloride, indicating caveolae-mediated pathway and macropinocytosis both contributed to the endocytosis of AN-75 (Supplementary Fig. 12a). In contrast, treatment with these three inhibitors markedly decreased the cellular uptake of AN-95, suggesting that AN-95 was likely internalized into mesangial cells through macropinocytosis, caveolae- and clathrin-mediated endocytosis (Supplementary Fig. 12b). The uptake of AN-130 was significantly decreased only with amiloride, indicating a macropinocytosis-mediated pathway (Supplementary Fig. 12c). In addition, cells treated with each inhibitor caused a more obvious reduction in

cellular uptake of AN-95 than that of AN-75 or AN-130, as indicated by 32.5%, 32.6%, or 44.2% inhibition in nystatin, chlorpromazine, or amiloride for AN-95, respectively (Supplementary Fig. 12d–f).

Above all, in vivo cell-specific accumulation of ANs might be associated with the size-dependent internalization of ANs by mesangial cells. AN-95 exhibited the greatest potential of internalization in mesangial cells via macropinocytosis, caveolae- and clathrin-mediated endocytosis. This might explain why the maximal mesangial cellular deposition was observed in AN-95. Thus, ANs with an average size distribution of ~ 95 nm was preferentially exploited for selective delivery of CLT to mesangial cells in the following studies.

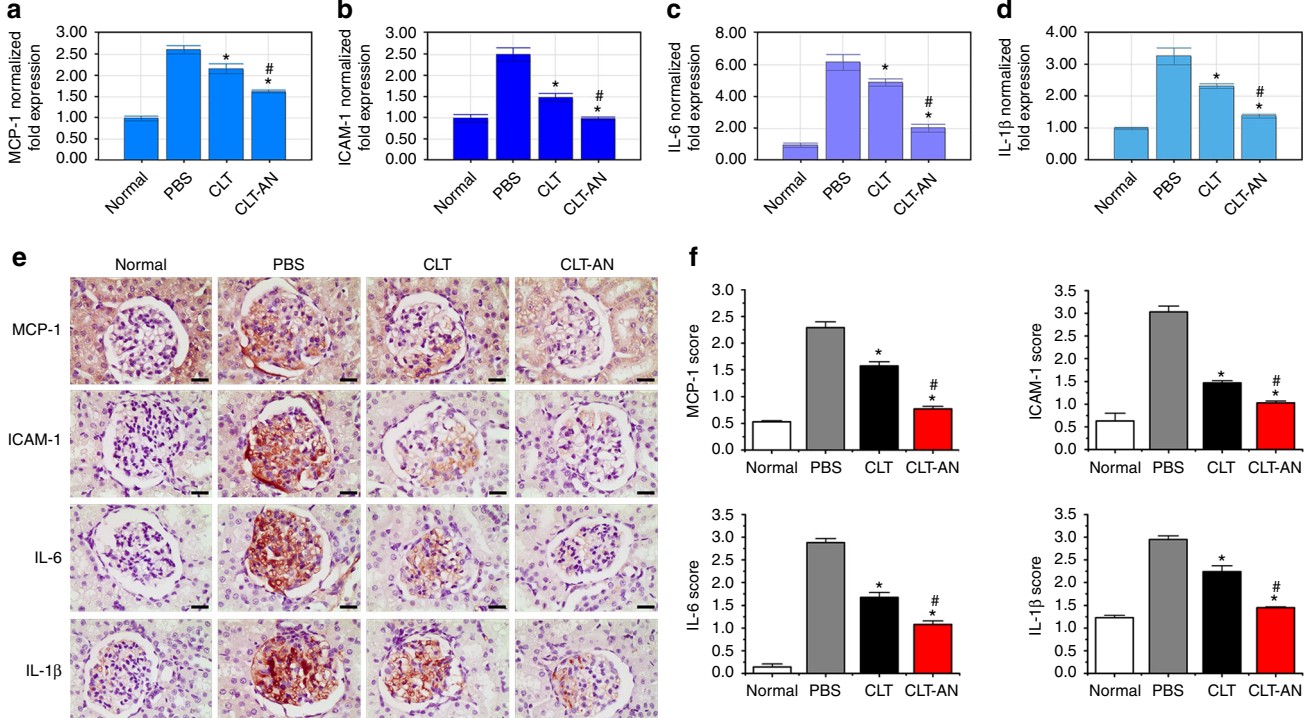

**Fig. 6** Anti-inflammatory effects of CLT and CLT-AN in anti-Thy1.1 nephritic rats. **a–d** Real-time PCR analysis of renal mRNA levels of MCP-1 (**a**), ICAM-1 (**b**), IL-6 (**c**), IL-1β (**d**) in anti-Thy1.1 nephritic rats on day 1 after treatment with CLT or CLT-AN. Data are shown as normalized fold expressions relative to normal group using β-actin mRNA as internal control. **e** Representative photomicrographs of immunostaining for MCP-1, ICAM-1, IL-6, and IL-1β in kidney tissue sections taken from anti-Thy1.1 nephritic rats on day 1 after treatment with CLT or CLT-AN. *Scale bars*, 20 μm. **f** The levels of MCP-1, ICAM-1, IL-6, and IL-1β were semiquantitatively scored as described in Methods on the basis of immunohistochemical results. In panels **a–d** and **f** data are mean ± s.d. (*n* = 5), results are representative of two independent experiments. *$P < 0.05$ vs. PBS group; #$P < 0.05$ vs. CLT group. Statistical significance was determined by one-way ANOVA with Tukey post hoc test

**CLT-AN improves the efficacy of CLT in anti-Thy1.1 nephritis.** By rationally tailoring the formulation and preparation conditions, we encapsulated CLT into ANs to afford nanoparticles with an average size distribution of ~ 95 nm (CLT-AN). Detailed characterizations of CLT-AN were provided in the Supplementary Fig. 13a–d. Next, an in vivo glomeruli biodistribution study was performed to examine the ability CLT-AN to deliver CLT to mesangial cells. CLT delivered as CLT-AN accumulated in the glomeruli to a significantly greater extent than free CLT at all time points (Supplementary Fig. 13e), which showed a much greater area under the curve ($AUC_{0-4 h}$) and maximal concentration ($C_{max}$) (Supplementary Table 1). We have confirmed the mesangial cellular distribution of AN-95 as described above (Fig. 3e). Collectively, these results suggest the excellent mesangial cells-targetability of CLT-AN and its potential for improving the therapeutic efficacy of CLT.

To examine the therapeutic efficacy of CLT-AN, we investigated the ability of early (treatment initiation on day 0) or delayed (day 2) CLT-AN treatment to reduce proteinuria and acute glomerular pathology in the reversible anti-Thy1.1 nephritis (Fig. 4; Supplementary Figs. 14–17). As expected, CLT-AN treatment led to significantly lower urinary protein excretion than treatment with free CLT regardless of when it was initiated (Fig. 4a, e). Also, CLT-AN (either early or delayed treatment) markedly attenuated both mesangial hypercellularity and ECM expansion (Fig. 4b–d, f–h). Moreover, no significant differences were observed in total glomerular cellularity and mesangial matrix score between the early CLT-AN-treated group and the normal control (Fig. 4c, d).

The immunostaining against ED-1, α-SMA, and PCNA showed that CLT-AN (either early or delayed treatment) markedly inhibited the glomerular invasion of monocytes/macrophages and mesangial cells activation/proliferation (Supplementary Figs. 14a and 15a). Through semiquantitative analysis, animals treated with CLT-AN showed fewer ED-1-positive cells per glomerular cross-section, and lower α-SMA score and fewer PCNA-positive cells per glomerular cross-section compared to animals treated with free CLT (Supplementary Figs. 14b–d and 15b–d). The immunostaining against Col I, Col IV, and FN-1 further showed CLT-AN treatment significantly reduced ECM accumulation (Supplementary Figs. 16a and 17a). CLT-AN treatment resulted in the lower ECM protein scores than treatment with free CLT (Supplementary Figs. 16b–d and 17b–d). The empty albumin nanoparticles (empty-AN) treatment exhibited no impacts on the level of proteinuria and acute glomerular lesions in reversible anti-Thy1.1 nephritis (Supplementary Fig. 18).

Moreover, we examined the efficacy of delayed CLT-AN treatment on proteinuria and progressive glomerular lesions in the irreversible anti-Thy1.1 nephritis. As expected, CLT-AN significantly improved the therapeutic efficacy of free CLT on proteinuria, infiltration of monocytes/macrophages, mesangial cell proliferation, and ECM expansion in anti-Thy1.1 nephritic rats on day 14 after disease induction (Fig. 5; Supplementary Figs. 19 and 20). Also, empty-AN showed no effects on proteinuria and progressive glomerular lesions in irreversible anti-Thy1.1 nephritis rats (Supplementary Fig. 21).

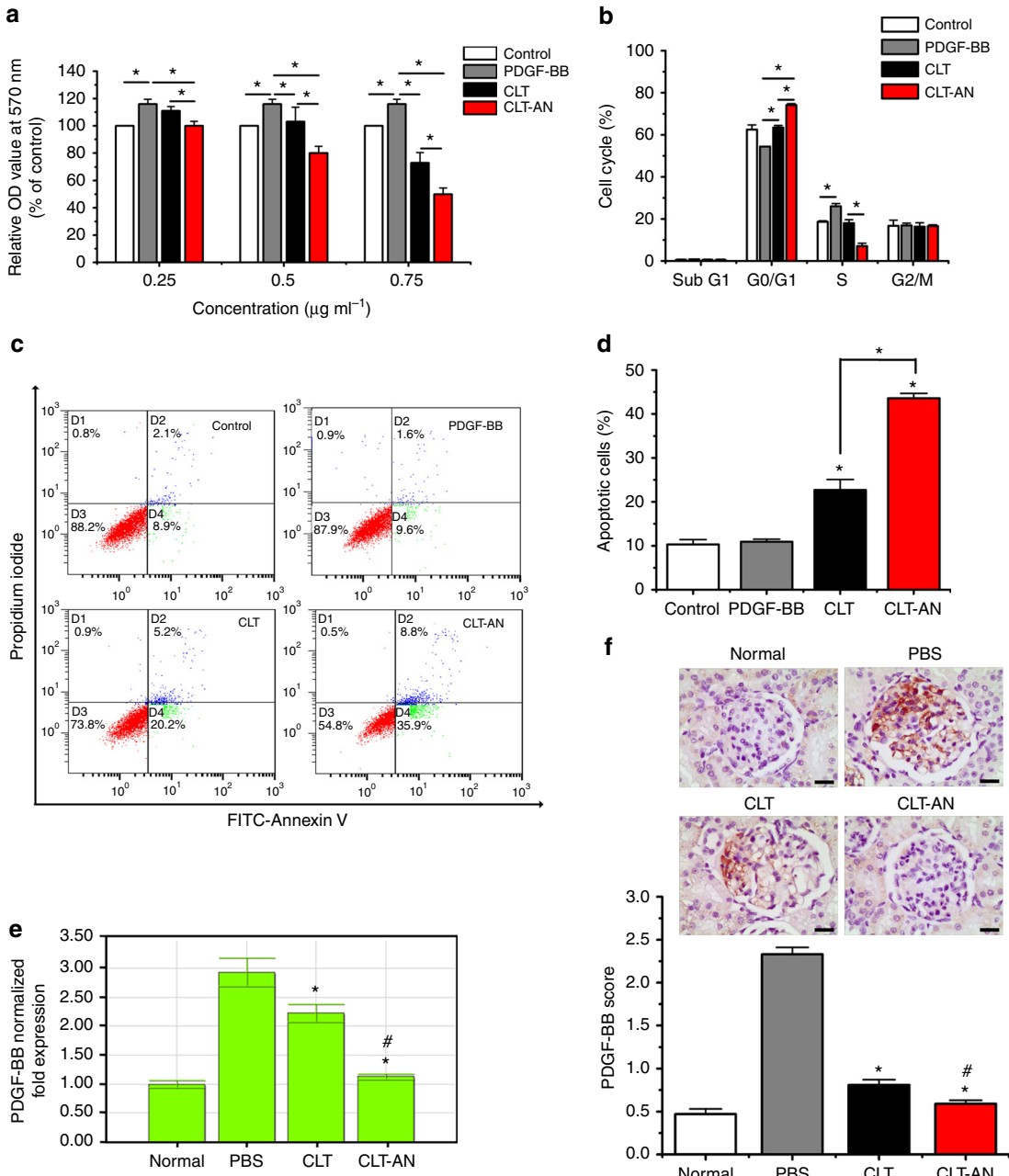

**Fig. 7** Anti-proliferative effects of CLT and CLT-AN in vitro and in vivo. **a** Effects of CLT and CLT-AN on PDGF-BB-induced proliferation of HBZY-1 cells. **b** Flow cytometry analysis of the effects of CLT and CLT-AN on the cell cycle distribution of HBZY-1 cells in the presence of PDGF-BB ($C_{CLT} = 0.25\ \mu g\ mL^{-1}$). **c** Representative quadrant plot obtained by flow cytometry analysis showing the ability of CLT and CLT-AN to induce apoptosis in HBZY-1 cells in the presence of PDGF-BB. **d** Flow cytometry analysis of the proportions of apoptotic HBZY-1 cells after 24-h exposure to CLT or CLT-AN in the presence of PDGF-BB ($C_{CLT} = 0.5\ \mu g\ mL^{-1}$). In panels **a**, **b**, and **d** data are mean $\pm$ s.d. ($n = 3$), results are representative of three independent experiments. *$P < 0.05$. Statistical significance was determined by one-way ANOVA with Tukey post hoc test. **e** Real-time PCR analysis of renal mRNA levels of PDGF-BB in anti-Thy1.1 nephritic rats on day 5 after treatment with CLT or CLT-AN. Data are shown as normalized fold expressions relative to normal group using β-actin mRNA as internal control. **f** Representative photomicrographs of immunostaining for PDGF-BB in kidney tissue sections from anti-Thy1.1 nephritic rats on day 5 after treatment with CLT or CLT-AN, and the levels of PDGF-BB were semiquantitatively scored as described in Methods on the basis of immunochemical results. *Scale bars*, 20 μm. In panels **e**, **f** data are mean $\pm$ s.d. ($n = 5$), results are representative of two independent experiments. *$P < 0.05$ vs. PBS group; #$P < 0.05$ vs. CLT group. Statistical significance was determined by one-way ANOVA with Tukey post hoc test

Taken together, CLT-AN with an optimal size of 95-nm selectively accumulated at mesangial cells thus forming a reservoir to display sustained therapeutic effects. CLT-AN showed better capacity than CLT in reducing proteinuria, ameliorating the acute and progressive glomerular lesions in anti-Thy1.1 nephritic rats through anti-inflammatory, anti-proliferative, and anti-fibrotic effects.

**Anti-inflammatory mechanisms of CLT and CLT-AN.** Previous studies have demonstrated the monocyte chemoattractant protein-1 (MCP-1) and intercellular adhesion molecule-1 (ICAM-1) secreted by mesangial cells are two well-known contributors to the recruitment of monocytes into glomeruli[35, 36]. These infiltrating cells further produce proinflammatory cytokines such as interleukin-6 (IL-6) and interleukin-1β (IL-1β),

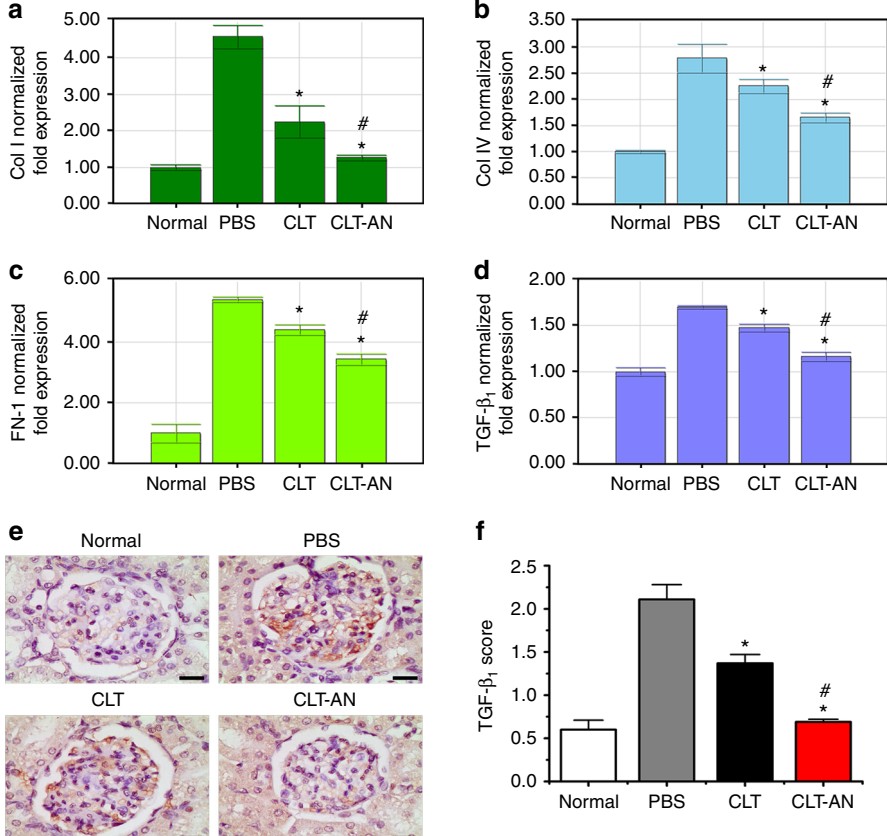

**Fig. 8** Anti-fibrotic effects of CLT and CLT-AN in anti-Thy1.1 nephritic rats. **a–d** Real-time PCR analysis of renal mRNA levels of Col I (**a**), Col IV (**b**), FN-1 (**c**), TGF-$\beta_1$ (**d**) in anti-Thy1.1 nephritic rats on day 5 after treatment with CLT or CLT-AN. Data are shown as normalized fold expressions relative to normal group using $\beta$-actin mRNA as internal control. **e** Representative photomicrographs of immunostaining for TGF-$\beta_1$ in kidney tissue sections taken from anti-Thy1.1 nephritic rats on day 5 after treatment with CLT or CLT-AN. *Scale bars*, 20 μm. **f** The levels of TGF-$\beta_1$ were semiquantitatively scored as described in Methods on the basis of immunochemical results. In panels **a–d**, **f** data are mean ± s.d. ($n = 5$), results are representative of two independent experiments. *$P < 0.05$ vs. PBS group; #$P < 0.05$ vs. CLT group. Statistical significance was determined by one-way ANOVA with Tukey post hoc test

acting on mesangial cells and perpetuating the pathogenic cellular response[37, 38]. CLT-AN and free CLT attenuated the upregulation of MCP-1, ICAM-1, IL-6, and IL-1β at the mRNA and protein levels on day 1 after disease induction, and the efficacy of CLT-AN was greater than that of CLT (Fig. 6). These findings suggest CLT and CLT-AN may inhibit monocyte infiltration into glomeruli via multiple pathways. Firstly, they may directly block monocytes infiltration by reducing the mRNA levels of both MCP-1 and ICAM-1. Secondly, they prevented IL-6 and IL-1β from stimulating mesangial cells to release harmful factors by decreasing the levels of IL-6 and IL-1β. In contrast to CLT treatment, CLT-AN was proven more potent in reducing levels of these inflammatory mediators in nephritic rats, leading to superior anti-inflammatory effects.

Nuclear factor kappa B (NF-κB) plays an important role in cytokine production and inflammatory cell recruitment[39]. In the anti-Thy1.1 nephritic rat model, OX7 bound to Thy1.1 antigens present on mesangial cells thus forming immune complexes, which stimulated NF-κB to translocate from the cytoplasm to the nucleus and activate transcription of pro-inflammatory mediator genes including *MCP-1, ICAM-1, IL-6, and IL-1β*[39]. CLT has been proven with the potent ability to inhibit NF-κB signaling pathway[9]. Here, we also confirmed the inhibitory effect of CLT or CLT-AN on NF-κB expression in nephritic rats at day 1 after disease induction based on immunohistochemistry analysis of kidney tissue sections (Supplementary Fig. 22). Thus, at the early

stage of anti-Thy1.1 nephritis, CLT or CLT-AN inhibited monocyte infiltration into the glomeruli via suppression of pro-inflammatory mediators, which may result from the inhibition of NF-κB signaling pathway.

**Anti-proliferative mechanisms of CLT and CLT-AN.** Platelet-derived growth factor-BB (PDGF-BB) as a potent mitogen plays critical role in the progression of GN[40, 41]. By an autocrine mechanism, mesangial cells and infiltrating monocytes secrete abundant PDGF-BB during the progression of renal injury, which stimulates proliferation of mesangial cells. PDGF-BB treatment significantly increased the proliferation of HBZY-1 cells, which was inhibited in a dose-dependent manner by CLT or CLT-AN, moreover, the anti-proliferative effect was greater with CLT-AN (Fig. 7a).

Next, we examined the effects of CLT and CLT-AN on the cell cycle distribution of HBZY-1 cells in the presence of PDGF-BB (Fig. 7b; Supplementary Fig. 23a, b). PDGF-BB induced more cells to enter S phase, with a concomitant decrease in the number of cells in G0/G1 phase. Conversely, CLT or CLT-AN induced G0/G1 arrest, significantly increasing the number of cells in G0/G1 phase and reducing the number of cells in S phase. These effects were greater with CLT-AN at a dose of 0.25 μg mL$^{-1}$ than with CLT at the same dose (Fig. 7b). At concentrations above 0.25 μg mL$^{-1}$, CLT treatment led to a larger number of cells in G0/G1 phase but a smaller number of cells in sub-G1 phase than

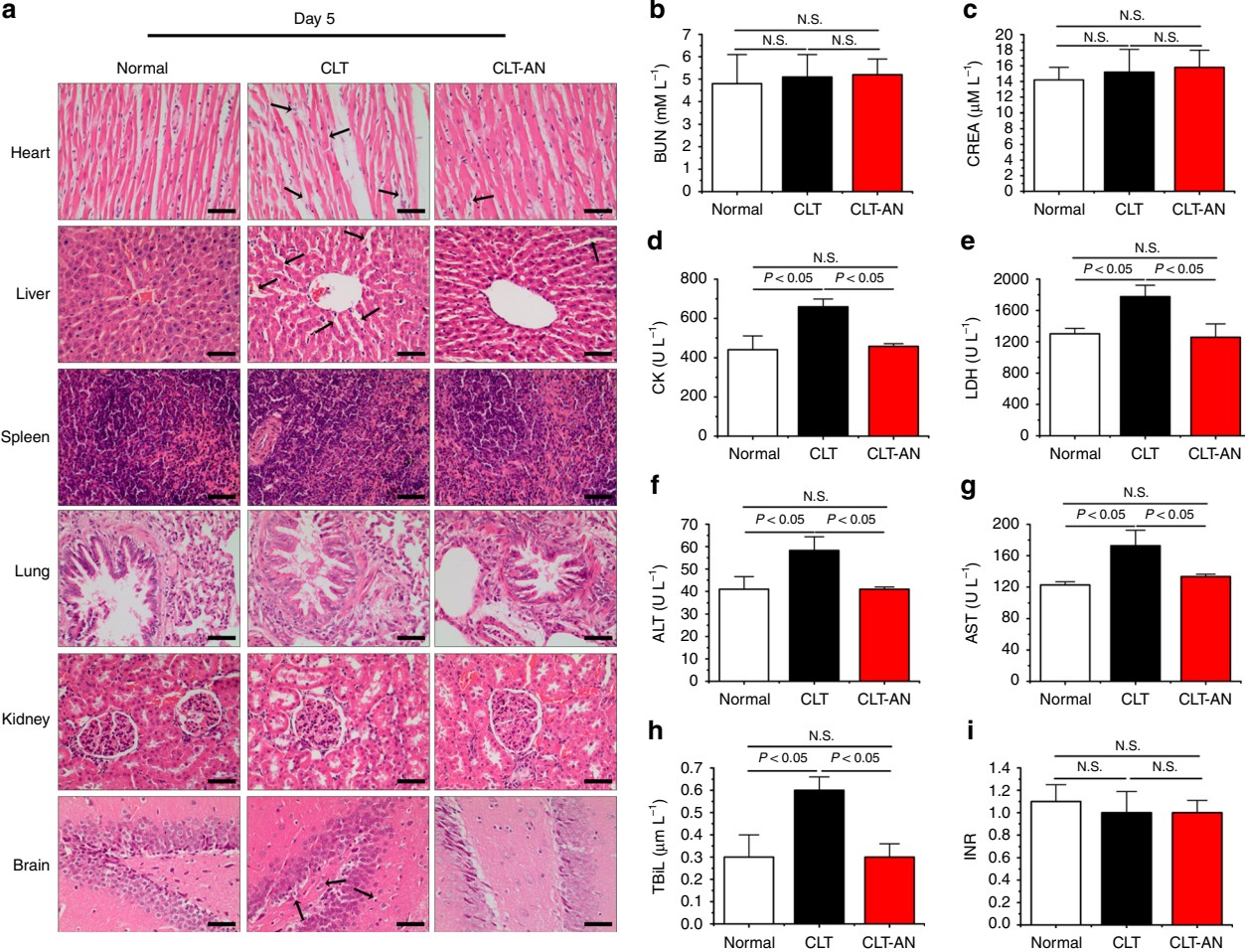

**Fig. 9** CLT-AN reduces systemic toxicity of CLT in rats. **a** Hematoxylin & eosin (H&E) staining assay of heart, liver, spleen, lung, kidney, and brain on day 5 after CLT or CLT-AN treatment. *Black arrows* in heart samples indicate atrophy of myocardial cells and myofibrillar loss; *arrows* in liver samples indicate atrophy of hepatic cells and dilatation of blood sinus; and *arrows* in brain samples indicate pyknosis of neuron and red neurons. *Scale bars*, 50 μm. **b–i** BUN, CREA, CK, LDH, ALT, AST, TBiL, and INR levels on day 5 after CLT or CLT-AN treatment. Data are mean ± s.d. (*n* = 5), results are representative of two independent experiments. Statistical significance was determined by one-way ANOVA with Tukey post hoc test. N.S., not significant

CLT-AN treatment (Supplementary Fig. 23a, b). The sub-G1 phase, when cells have hypodiploid DNA content, is thought to be a hallmark of apoptosis[42].

We then quantitated the extent of apoptosis induced by CLT or CLT-AN. Rate of apoptosis was significantly higher in HBZY-1 cells treated with CLT or CLT-AN than in cells treated with cell medium alone or PDGF-BB alone (Fig. 7c, d). The apoptotic rate was approximately twofold higher with CLT-AN than with CLT (Fig. 7d). No necrosis was observed in cells treated with CLT or CLT-AN. These data indicate that CLT or CLT-AN could not only arrest the mesangial cells at G0/G1 phase but also induce cells to undergo apoptosis. Herein, more cell apoptosis induction and cell cycle arrest by CLT-AN in contrast to free CLT. HBZY-1 cells took up significantly more CLT-AN than free CLT at all concentrations and given time points (Supplementary Fig. 23c, d), suggesting that the drug is taken up by cells much more efficiently when encapsulated in nanoparticles.

To examine whether CLT or CLT-AN might inhibit mesangial cells proliferation by directly affecting PDGF-BB levels, we evaluated the effects of CLT or CLT-AN on the level of PDGF-BB mRNA and protein in anti-Thy1.1 nephritic rats. The mRNA level of PDGF-BB was upregulated on day 5 after disease induction. CLT or CLT-AN treatment significantly attenuated

this upregulation, and the efficacy of CLT-AN was greater than that of CLT (Fig. 7e). Similar results were observed based on immunohistochemistry analysis of kidney tissue sections (Fig. 7f).

Collectively, the pharmacological mechanism of CLT or CLT-AN inhibiting mesangial cells proliferation in anti-Thy1.1 nephritic rats might involve the inhibition of PDGF-BB production, the reduction in numbers of proliferative mesangial cells resulting from the cell cycle arrest and cell apoptosis induction by CLT or CLT-AN. Moreover, CLT-AN showed the superior anti-proliferative effects to free CLT.

**Anti-fibrotic mechanisms of CLT and CLT-AN.** ECM over-production by mesangial cells is associated with progression of glomerulofibrosis and sclerosis, and transforming growth factor-$\beta_1$ (TGF-$\beta_1$) is a critical mediator of ECM deposition in many glomerular diseases[43]. The induction of anti-Thy1.1 nephritis upregulated the mRNA levels of Col I, Col IV, FN-1, and TGF-$\beta_1$ on day 5 (Fig. 8a–d), which were downregulated by CLT or CLT-AN, while CLT-AN attenuated the upregulated genes to a greater extent than CLT. Similar results were observed based on immunohistochemistry analysis of kidney tissue sections (Fig. 8e, f; Supplementary Fig. 16). Thus, CLT and CLT-AN inhibited

ECM expansion in anti-Thy1.1 nephritis by inhibiting mesangial cell proliferation and reducing ECM protein secretion. This reduction of ECM proteins may occur via direct downregulation of Col I, Col IV, and FN-1 mRNA, as well as indirect downregulation of TGF-$\beta_1$ mRNA. Overall, CLT-AN showed better anti-fibrotic effects than CLT.

**CLT-AN reduces systemic toxicity of CLT in rats.** We investigated the systemic toxicity of CLT or CLT-AN in rats on day 1, 5, 14, and 21 after intravenous administration (Fig. 9; Supplementary Figs. 24–26). Animals exposed to CLT or CLT-AN showed no visible lesions in the spleen, lung, or kidney at all time points (Fig. 9a; Supplementary Figs. 24, 25a, and 26a). Also, neither CLT nor CLT-AN significantly altered levels of BUN or CREA (Fig. 9b, c; Supplementary Figs. 25b, c and 26b, c).

CLT at 1 mg kg$^{-1}$ induced serious pyknosis of neuron and red neurons in the brain on day 1, 5, or 14 (Fig. 9a; Supplementary Figs. 24 and 25a), and even liquefactive necrosis foci on day 21 (Supplementary Fig. 26a). In contrast, animals treated with CLT-AN showed no obvious signs of neurotoxicity. CLT also caused severe atrophy of myocardial cells and myofibrillar loss in the heart (Fig. 9a; Supplementary Figs. 24, 25a, and 26a). Consistently, CLT increased the levels of CK or LDH (Fig. 9d, e; Supplementary Figs. 25d, e and 26d, e). However, CLT-AN showed slight cardiotoxicity with increased levels of LDH on day 21 without significantly altering the levels of CK or LDH on day 5 or 14 (Fig. 9d, e; Supplementary Figs. 25d, e and 26d, e). In addition, CLT resulted in severe dilatation of blood sinus and loss of hepatic cords in the liver on day 1 or 5 (Fig. 9a; Supplementary Fig. 24), and even diffuse edema of hepatic cells on day 14 or 21 (Supplementary Figs. 25a and 26a). In contrast, CLT-AN significantly reduced the hepatotoxicity at all given time points. CLT treatment elevated ALT, AST, TBiL levels on day 5, 14 or 21 (Fig. 9f–h, Supplementary Figs. 25f–h and 26f–h), while no significant differences in ALT, AST, TBiL levels were observed between CLT-AN-treated group and the normal group on day 5 (Fig. 9f–h). On day 21 post administration, CLT-AN treatment displayed elevated ALT, AST and TBiL levels, which were lower than the levels in free CLT-treated group (Supplementary Fig. 26f–h). Neither CLT nor CLT-AN significantly altered the INR levels at all tested time points (Fig. 9i, Supplementary Figs. 25i and 26i), suggesting the liver synthetic function was not influenced by CLT or CLT-AN.

Collectively, CLT-AN was relatively well-tolerated and that encapsulation of CLT in ANs significantly reduced the toxicity of CLT to major organs. The empty-AN treatment showed no visible signs of organ damage in rats (Supplementary Fig. 27a, b), and no significant alterations in all tested biochemical indicators on day 21 (Supplementary Fig. 27c–j).

Next, we examined whether the improved safety of CLT-AN was due to a more selective biodistribution. The biodistribution study demonstrated that drug concentration and $AUC_{0-4\,h}$ in heart or brain were significantly lower for CLT-AN than for free CLT (Supplementary Fig. 28a–d; Supplementary Table 2), which likely accounted for the alleviating lesions of CLT-AN in these organs. However, CLT-AN proven with reduced hepatotoxicity was observed with higher drug concentration and $AUC_{0-4\,h}$ in liver than that of free CLT (Supplementary Fig. 28a–d; Supplementary Table 2). Nanoparticles free of ligand-modification and with sizes from 10 to 200 nm can easily be captured by Kupffer cells of reticuloendothelial system (RES), and the high-negative surface charge of nanoparticles may accelerate the uptake by Kupffer cells via scavenger receptors[44, 45]. CLT-AN may reach the liver via non-specific phagocytosis of RES, and accumulate in the Kupffer cells rather than hepatocytes.

To prove this hypothesis, FITC-labeled CLT-AN (FITC-CLT-AN) was used to visualize the distribution profile of CLT-AN in the liver. The introduction of FITC to CLT-AN formulation did not alter average particle size and zeta potential of CLT-AN ($95.33 \pm 0.17$ nm; $-22.8 \pm 0.1$ mV) (Supplementary Fig. 29a, b). Five minutes after intravenous administration of FITC-CLT-AN in rats, livers were harvested. Immunofluorescence staining against CD68, a macrophage marker[46], was used to examine the accumulation of CLT-AN in Kupffer cells. The co-localization of FITC-CLT-AN with CD68 confirmed CLT-AN were largely retained in Kuffer cells rather than in hepatocytes (Supplementary Fig. 29c). Accordingly, the limited drug distribution of CLT-AN in the hepatocytes might explain their reduced hepatotoxicity.

## Discussion

MsPGN such as IgA nephropathy often leads to end-stage renal failure[47]. Currently, treatment remains controversial because none of the current treatment regimens produce a convincing benefit[47]. The search for agents favorably modifying mesangial cell disorders are of pivotal clinical importance. Anti-Thy1.1 nephritis is a well-characterized experimental model of immune complex-mediated MsPGN such as IgA nephropathy[48]. Induced by a single injection of anti-thymocyte serum or anti-Thy 1.1 monoclonal antibody, the model is characterized by an early phase of inflammatory cell infiltration into glomeruli and complement-dependent mesangiolysis followed by marked mesangial proliferation and excessive ECM deposition[12, 48]. However, the glomerular lesions and proteinuria in this model are reversible with spontaneous resolution in 2 to 4 weeks after anti-Thy1.1 antibody administration[12]. In contrast, uninephrectomized rats treated with a single injection of the anti-Thy1.1 antibody produces an irreversible model of anti-Thy1.1 nephritis for evaluating the effects of interventions on progressive proteinuria and mesangial lesions in MsPGN[12, 48].

In this study, CLT showed potent effects against proteinuria, inflammation, glomerular hypercellularity, and ECM deposition in anti-Thy1.1 nephritis through anti-inflammatory, anti-proliferative, and anti-fibrotic effects, representing a promising agent for the treatment of MsPGN such as IgA nephropathy. However, CLT treatment induced severe cardiotoxicity, hepatotoxicity, and neurotoxicity in rats, as reflected by the atrophy of myocardial cells and myofibrillar loss in the heart, atrophy, or edema of hepatic cells in the liver, and pyknosis of neuron and red neurons or liquefactive necrosis foci in the brain. Despite the efficacy of CLT against proteinuria and glomerular lesions, its potential toxicity remains unresolved.

From a pharmaceutical point of view, targeted delivery of CLT to mesangial cells may help improve the efficacy yet reducing systemic toxicity. Here, we selected ANs as the carrier system mainly due to following reasons: (i) ANs made from HSA are biodegradable, non-toxic and non-immunogenic, which can facilitate rapid translation into the clinic; (ii) ANs can be prepared by a facile and highly reproducible method; (iii) hydrophobic drugs such as CLT can easily be encapsulated into ANs due to the high-binding affinity of CLT to albumin[49]. The ANs fabricated by HSA achieved selective delivery to the mesangial cells in a size-dependent manner. ANs of 95-nm maximized the uptake by mesangial cells via caveolae-, clathrin-mediated pathway, and macropinocytosis. CLT-loaded ANs with an average diameter of ~ 95 nm selectively addressed high CLT concentration to mesangial cells and prolonged the drug availability at this local site of action. The potential mechanisms of targeted localization of CLT-AN in mesangial cells are proposed as follows: (i) the great blood flow rate in the kidney together with the high

hydraulic pressure in glomerular capillary[50], might favor CLT-AN retaining in glomeruli; (ii) fenestrations of 80–150 nm of endothelium between glomerular capillaries and mesangium, might allow CLT-AN to permeate across the endothelial fenestrations and gain access to mesangial cells[51]. Meanwhile, 10 nm-size-cutoff of glomerular filtration barrier can prevent CLT-AN from filtering into Bowman's capsule space[51]; (iii) Further, CLT-AN might be internalized into mesangial cells by phagocytosis[52]. Compared with free CLT, targeted delivery of CLT by CLT-AN to mesangial cells significantly enhanced the therapeutic efficacy of CLT in anti-Thy1.1 nephritis rats. In addition, CLT-AN altered the biodistribution of CLT with reduced drug accumulation in heart, brain, and liver, thereby minimizing CLT-related off-target toxicity. Taken together, encapsulating CLT in ANs represents a promising strategy for increasing the therapeutic index and reducing the systemic toxicity of CLT for treating MsPGN and related glomerular diseases.

## Methods

**Materials**. Celastrol (CLT) and glycyrrhetinic acid (98.0% pure) were purchased from Chengdu Must Biotechnology (Chengdu, China). MPA (98.0% pure) was obtained from Hubei Hengshuo Chemical (Wuhan, China). HSA was a gift from Sichuan Yuanda Shuyang Pharmaceutical (Chengdu, China). Soybean oil was provided by Beiya Medical Oil (Tieling, China). PDGF-BB was supplied by R&D Systems (Minneapolis, USA). FITC, 3-(4,5-dimethylthiazol-2-yl)-2,5-diphenlte-trazolium bromide (MTT), propidium iodide (PI), and endocytosis inhibitors including nystatin, amiloride, and chlorpromazine were purchased from Sigma-Aldrich (USA). All other chemical reagents and solvents were of analytic grade or above.

**Animals and cell culture**. Male Kunming mice (4–5 weeks) and male Sprague-Dawley rats (5–6 weeks) were purchased from Dashuo Biotechnology (Chengdu, China) and maintained under standard housing conditions. All animal care and experiments were conducted in compliance with the requirements of the National Act on the use of experimental animals (China) and were approved by the Institutional Animal Care and Ethic Committee of Sichuan University (Approved No. SYXK2013-113). Male rats or mice that were randomly assigned to treatment groups after ensuring that the randomization would allow body weight to be matched between the groups prior to the start of the experiment.

HBZY-1 rat glomerular mesangial cells were purchased from the China Center for Type Culture Collection, and maintained in DMEM with low glucose (GIBCO, USA) supplemented with 10% (v/v) fetal bovine serum, streptomycin (50 U mL$^{-1}$) and penicillin (50 U mL$^{-1}$). Cells were cultured at 37 °C in a humidified atmosphere containing 5% $CO_2$. The cells showed the correct phenotype, with the expected morphology and growth curves. The mycoplasma detecting kit (catalogue no. MD001), purchased from Shanghai Yise Medical Technology was performed to test mycoplasma contamination in the cells.

We used sample sizes of five for animal experiments, and of three for cell experiments. No data or animals were excluded due to being outliers.

**Animal models and therapeutic experiments**. The reversible and irreversible models of anti-Thy1.1 nephritis are widely used experimental animal models that simulate human MsPGN[12]. A reversible rat model of mesangial proliferative glomerulonephritis (MsPGN) was induced on day 0 by tail vein injection of anti-Thy1.1 monoclonal antibody (1 mg kg$^{-1}$, OX7, MCA47EL, Serotec) to male Sprague-Dawley rats (5–6 weeks) following a previously established method[53]. Since binding of OX7 and Thy1.1 antigen onto mesangial cells reaches maximal at 1 h after injection[54], early drug treatments were initiated 2 h after injection to avoid binding and deposition of OX7. The early mesangiolysis at day 1 is followed by marked mesangial cell proliferation and expansion of ECM starting at day 2 (Supplementary Fig. 3), thus the delayed treatments were initiated at day 2 after disease induction. MPA, as a beneficial agent against anti-Thy1.1 nephritis[13, 14], was selected as the standard treatment control. For an irreversible model of MsPGN, rats were unilaterally nephrectomized and 6 days later intravenously injected with the dose of 1 mg kg$^{-1}$ OX7[12]. Also, treatments in this model were initiated at day 2 after disease induction.

For the early treatment of CLT in the reversible model of anti-Thy1.1 nephritis, 60 rats were randomized into six groups: normal group, which were healthy rats treated with PBS; PBS group, which were nephritic rats treated with PBS; MPA group, which were nephritic rats treated with 30 mg kg$^{-1}$ MPA; and three CLT groups, which were nephritic rats treated with 1 mg kg$^{-1}$ CLT (LD-CLT), 2 mg kg$^{-1}$ CLT (MD-CLT) or 3 mg kg$^{-1}$ CLT (HD-CLT). PBS or drugs were administered to rats via tail vein injection once every day until nephrectomy. In this rat MsPGN model, the inflammatory cell infiltration peaks on day 1 after disease induction,

and maximal glomerular cell proliferation and ECM expansion occur on day 5[55]. Therefore, five rats in each group were sacrificed on day 1 for evaluation of anti-inflammatory effects, and the remaining rats were sacrificed on day 5 for analysis of anti-proliferative and anti-fibrotic effects. For the delayed treatment of CLT in the reversible model of anti-Thy1.1 nephritis, 30 rats were randomized into six groups as described above, and all animals were sacrificed on day 5. The induction of reversible anti-Thy1.1 nephritis and treatment regimens of CLT or MPA were schematically presented in Fig. 1e and Supplementary Fig. 4e.

For the treatment of CLT in the irreversible model of anti-Thy1.1 nephritis, 35 rats were randomized into seven groups: 2-K Control group, which were nonnephrectomized two-kidney rats treated with PBS; 1-K Control group, which were uninephrectomized one-kidney rats treated with PBS; PBS group, which were nephritic rats treated with PBS; MPA group, which were nephritic rats treated with 30 mg kg$^{-1}$ MPA; and three CLT groups, which were nephritic rats treated with 1 mg kg$^{-1}$ CLT (LD-CLT), 2 mg kg$^{-1}$ CLT (MD-CLT), or 3 mg kg$^{-1}$ CLT (HD-CLT). PBS or drugs were administered to rats via tail vein injection once every other day until nephrectomy. All animals were sacrificed on days 14 after disease induction. In all, 2-K Control and 1-K Control served as controls. The induction of irreversible anti-Thy1.1 nephritis and treatment regimens of CLT or MPA were schematically presented in Supplementary Fig. 7e.

To study the therapeutic effect of CLT-AN in the reversible model of anti-Thy1.1 nephritis, rats were randomized into five groups: normal group, which were healthy rats treated with PBS; PBS group, which were nephritic rats treated with PBS; empty-AN group, which were nephritic rats treated with empty-AN; CLT group, which were nephritic rats treated with free CLT (1 mg kg$^{-1}$); and CLT-AN group, which were nephritic rats treated with CLT-AN (1 mg kg$^{-1}$). For the early treatment, PBS, empty-AN or drugs were administered via tail vein injection once every other day until nephrectomy. For the delayed treatment, PBS or drugs were administered to rats once every day until nephrectomy. Animals were sacrificed on days 1 and 5 as described in studies of therapeutic effects of CLT. The induction of reversible anti-Thy1.1 nephritis and treatment regimens of CLT or CLT-AN were schematically presented in Fig. 4i.

To study the therapeutic effect of CLT-AN in the irreversible model of anti-Thy1.1 nephritis, rats were randomized into six groups: In all, 2-K Control group, which were two-kidney nonnephrectomized rats treated with PBS; 1-K Control group, which were one-kidney uninephrectomized rats treated with PBS; PBS group, which were nephritic rats treated with PBS; empty-AN group, which were nephritic rats treated with empty-AN; CLT group, which were nephritic rats treated with free CLT (1 mg kg$^{-1}$); and CLT-AN group, which were nephritic rats treated with CLT-AN (1 mg kg$^{-1}$). PBS, empty-AN or drugs were administered via tail vein injection once every other day until nephrectomy. All animals were sacrificed on days 14 after disease induction. 2-K Control and 1-K Control served as controls. The induction of irreversible anti-Thy1.1 nephritis and treatment regimens of CLT or CLT-AN were schematically presented in Fig. 5e.

**Proteinuria measurement**. Urine samples (24-h) were collected from all rats while they were housed individually in metabolic cages on day 5 after the reversible anti-Thy1.1 nephritis induction or on day 14 after the irreversible anti-Thy1.1 nephritis induction. Urinary protein concentration was measured using the Bradford Protein Assay Kit (Solarbio, Beijing, China).

**Renal glomerular histopathology**. Renal tissues taken from animals on day 5 or 14 were fixed in 10% formalin, and 4-μm paraffin sections were stained with PAS. Tissue sections were examined for glomerular histopathology and photographed under a light microscope (Olympus BX5, Tokyo, Japan). Glomerular hypercellularity was assessed on the basis of total cell number per glomerular cross-section (150 sections per group), which was counted using cellSens Standard digital imaging software (Olympus). ECM deposition in the mesangium was also investigated in glomerular cross sections (150 per group) and graded semiquantitatively on a scale of 0–4[56], where grade 0 indicates very weak or absent mesangial staining, and grades 1–4 indicate focally strong staining in, respectively, < 25%, 25–50, 50–75 or 75–100% of the glomerular tuft. Average ECM deposition scores were calculated for each animal group based on scores on the five-point scale and the corresponding numbers of glomeruli. All analyses were performed by one certified pathologists blinded to sample identity.

**Immunohistochemistry**. Immunohistochemical staining was performed as previously reported[57]. Briefly, paraffin-embedded kidney sections were deparaffinized, subjected to antigen retrieval and treated to block endogenous peroxidase activity. Sections were incubated for 45 min at 37 °C with primary antibodies against one of the following antigens: ED-1 (1:100; Abcam, ab31630), α-SMA (1:200; Abcam, ab7817), PCNA (1:300; Abcam, ab18197), PDGF-BB (1:200; Abcam, ab21234), MCP-1 (1:200; Abcam, ab25124), ICAM-1 (1:100; BD, 554967), IL-6 (1:50; Abcam, ab6672), IL-1β (1:100; Abcam, ab9722), NF-κB (1:50, Abcam, ab16502), Col I (1:500; Abcam, ab6308), Col IV (1:400; Abcam, ab19808), FN-1 (1:50; Abcam, ab2413), and TGF-β$_1$ (1:50; Abcam, ab92486). In negative controls, the primary antibody was replaced by buffer. After addition of 100 μL Agent A (horseradish peroxidase-conjugated ChemMate Envision reagent) from the anti-rabbit/rat general immunohistochemical test kit (Envision Detection Kit, GK500705), the color reaction was performed using 3,3-diaminobenzidine (Sigma-Aldrich, USA),

then each section was counterstained with hematoxylin. Tissues were examined for positive (yellowish-brown) staining and photographed under a light microscope (Olympus BX53, Tokyo, Japan). The representative photomicrographs of immunostaining for ED-1, MCP-1, ICAM-1, IL-6, IL-1β, NF-κB, α-SMA, PCNA, PDGF-BB, Col I, Col IV, FN-1, TGF-β₁ in kidney tissue sections taken from anti-Thy1.1 nephritic rats with the corresponding primary antibody or PBS were provided in Supplementary Fig. 30.

Glomerular monocyte/macrophage infiltration and glomerular cell proliferation were assessed by counting ED-1-positive cells on day 1, 5, or 14 and PCNA-positive cells on day 5 or 14 in 150 glomerular cross sections per animal group. These positive cells were counted using cellSens Standard digital imaging software (Olympus). Glomerular cross sections taken on day 5 or 14 were stained for α-SMA and the ECM proteins Col I, Col IV, and FN-1 (150 sections per group) to assess activation of mesangial cells and ECM deposition. The semiquantitative five-point grading system described above was used. The immunostaining for MCP-1, ICAM-1, IL-6, IL-1β, NF-κB, PDGF-BB, or TGF-β₁ was also evaluated and graded semiquantitatively in 150 glomerular cross sections per group based on the five-point grading system. All analyses were performed by one certified pathologists blinded to sample identity.

**Preparations and characterizations of FITC-ANs and CLT-AN**. FITC-labeled albumin nanoparticles (FITC-ANs) of different sizes (AN-75, AN-95, and AN-130) were prepared using an ultrasonication method. Briefly, predetermined amounts of soybean oil were dissolved in 2 mL of a mixture solvent of methylene chloride and ethyl acetate to provide the organic phase. Then, 1 mL of 20% (w/v) HSA and a small quantity of FITC labeled HSA (FITC-HSA) dispersed in 10 mL distilled water were added to the organic phase. The dispersion was intermittently ultrasonicated at 330 W for 8 min using a probe ultrasonicator (Xinzhi Biotechnology, Ningbo, China). The organic solvent was removed by vacuum rotary evaporation at 37 °C. FITC-HSA was conjugated as previously described[58]. Briefly, 10 mg of FITC and 2 mL of 20% (w/v) HSA were dissolved in carbonate buffer (0.05 M, pH 9.6) and stirred in dark for 9 h at room temperature. The solution was subsequently purified by dialysis in ultrapure water for 48 h, then FITC-HSA was obtained. To generate AN-75, AN-95 and AN-130, 10 mg, 25 mg or 45 mg of soybean oil was used in organic phase, respectively. As for CLT-loaded albumin nanoparticles (CLT-AN), 7 mg of CLT and 45 mg of soybean oil to dissolve in 2 mL of a mixture solvent of methylene chloride and ethyl acetate to provide the organic phase. Then, 1 mL of 20% (w/v) HSA dispersed in 10 mL distilled water were added to the organic phase. The dispersion was intermittently ultrasonicated at 490 W for 8 min. FITC-labeled CLT-AN (FITC-CLT-AN) was obtained by adding a small quantity of FITC-HSA into the aqueous phase of CLT-AN formulation.

Mean particle size and zeta potential of FITC-ANs, CLT-AN, and FITC-CLT-AN were determined in ultrapure water at 25 °C using dynamic light scattering on a Malvern Zetasizer (Malvern, NanoZS90, UK). Nanoparticle morphology was further examined using transmission electron microscopy (H-600, Hitachi, Japan) after staining samples with uranyl acetate.

Encapsulation efficiency (EE%) and drug loading efficiency (DLE%) of CLT-AN were measured using gel exclusion chromatography. In brief, a predetermined volume of freshly prepared CLT-AN was added into a Sephadex G50 column, on which CLT encapsulated in CLT-AN were separated from free CLT while eluting with distilled water. CLT content in the eluted CLT-AN ($W_e$) was measured using high-performance liquid chromatography (Agilent1260 Infinity, USA). Measurements were performed at 35 °C on a C18 column (5 μm, 150 × 4.6 mm, Kromasil). The mobile phase consisted of methanol and 10% acetate acid (95:5, v/v). The detection wavelength was 425 nm. An equal amount of nanoparticle suspension was subjected to similar procedures to quantify the total drug in the nanoparticle system ($W_t$). The total weight of an equivalent amount of dried nanoparticles ($W_d$) was also determined. EE% and DLE% were calculated using Equation (1) and Equation (2), respectively.

$$EE\% = W_e/W_t \times 100\% \qquad (1)$$

$$DLE\% = W_e/W_d \times 100\% \qquad (2)$$

**Glomerular mesangial cellular localization of ANs**. To examine the renal distribution profiles of FITC-ANs of different sizes, male Kunming mice were randomized to receive FITC-labeled AN-75, AN-95, or AN-130 (0.1 mg kg⁻¹ FITC) via tail vein injection (n = 3 per group). Control mice were injected with normal saline. Animals were sacrificed at 5 min after administration, and kidneys were excised and imaged using an ex vivo imaging system (Quick View 3000, Bio-Real, Austria). In our preliminary experiment, ANs achieved the maximal renal distribution at 5 min, thus the renal distribution profiles of FITC-ANs were compared at this time point. Ex vivo fluorescent images were analyzed semiquantitatively to generate statistical plots of average fluorescence intensity in kidney for each group.

In studies of the intrarenal accumulation of FITC-ANs, excised kidneys were snap-frozen in Tissue-Tek OCT compound (SAKURA, USA), and 4-μm sections were prepared, embedded with an antifading agent (Sigma) and observed under a confocal laser scanning microscope (CLSM, Olympus Fluoview FV 1000, USA). FITC-ANs deposition in glomeruli was analyzed semiquantitatively based on the intensity and spread of fluorescence, which was measured in terms of IOD[59] using cellSens Standard digital imaging software (Olympus). IOD in each treatment group was assessed for 150 randomly selected glomeruli in kidney sections, and investigators were blinded to animal treatment.

To examine the intraglomerular localization of FITC-ANs, excised kidneys were fixed with 10% formalin and further dehydrated using 15% (w/v) sucrose, followed by 30% (w/v) sucrose. Tissues were snap-frozen in OCT compound, and 4-μm frozen sections were prepared and stained with goat anti-α8 integrin antibody (1:100; R&D Systems, AF4076), followed by NL637 conjugated donkey anti-goat antibody (1:200; R&D Systems, NL002). Stained sections were observed by CLSM. Co-localization of FITC-ANs and α8-integrin was semi-quantitated using cellSens Standard digital imaging software (Olympus).

**Intracellular uptake pathways assay**. In the study of intracellular uptake pathways, HBZY-1 cells were seeded in six-well plates at a density of $3 \times 10^5$ cells per well in 2 mL of culture medium and allowed to attach in culture medium for 24 h. The effect of temperature on cellular uptake was performed at 4 °C and 37 °C to evaluate if ANs uptake occurs through an active process. To explore the probable mechanisms that ANs of different particle sizes deposited size-dependently in glomerular mesangial cells, endocytic pathways involved in mesangial cellular uptake of ANs of different particle sizes were studied. HBZY-1 cells were pre-incubated for 1 h at 37 °C with nystatin (10 μg mL⁻¹) to inhibit caveolae, amiloride (10 μg mL⁻¹) to inhibit macropinocytosis, or chlorpromazine (10 μg mL⁻¹) to inhibit the formation of clathrin vesicles, and then treating the cells with fluorescent AN-75, AN-95, and AN-130 for another 2 h. The concentrations of these endocytic inhibitors were not toxic to the cells in our preliminary experiment. Next, the cells were washed with 0.2 M ethanoic acid/0.5 M NaCl for three times to remove fluorescent ANs attached to the cell surface, then trypsinized, collected, washed, centrifuged, and resuspended in 0.3 mL PBS. The fluorescent intensity of cells was measured by a flow cytometer (Cytomics FC 500, Beckman Coulter, USA). The groups treated with fluorescent ANs but free of inhibitors treatment was used as controls, whereas the group without any treatment was served as a background in the flow cytometry analysis. Results were expressed as the relative uptake percentage compared to controls.

**Sample preparation and liquid chromatography–mass spectrometry (LC-MS)/MS analysis**. Cell lysate and glomerular lysate for LC-MS/MS analysis were prepared using a liquid-liquid extraction method. Aliquots (100 μL) of lysates were extracted using 1 mL of ethyl acetate after addition of the glycyrrhetinic acid IS (10 μL of 9.42 μg mL⁻¹ working solution). Extraction was performed by vortexing for 10 min and centrifuging at 12,000×g rpm for 5 min. An aliquot of supernatant (800 μL) was harvested, and the residue was mixed with 1 mL of ethyl acetate for a second extraction. The upper organic layers of both extractions were pooled and evaporated to dryness at 40 °C under a stream of nitrogen. The dried residue was reconstituted with 200 μL of mobile phase, vortexed for 1 min and centrifuged at 12,000×g rpm for 10 min. Finally, the supernatant was filtered through a 0.22-μm hydrophobic membrane for injection into the LC-MS/MS system.

In our preliminary experiment, we found that the liquid-liquid extraction method exhibited very low extraction recovery (about 10–15%) for tissue homogenates because of tight binding between CLT and tissue protein. Therefore, we developed a liquid–solid extraction method for tissues sample preparation. An aliquot (100 μL) of tissue homogenate and 10 μL of 4 μg mL⁻¹ glycyrrhetinic acid IS were mixed well in a miniature mortar. Then a predetermined amount of anhydrous sodium sulfate was added to absorb all the water in the mixture. The dried mixture was ground to a powder, which was suspended in 4 mL ethyl acetate, transferred into Eppendorf tubes and vortexed for 1 h. The liquid–solid suspension was then centrifuged at 12,000×g rpm for 5 min, and an aliquot (3 mL) of supernatant was evaporated at 40 °C under N₂. The dried residue was reconstituted with 200 μL of mobile phase, vortexed for 1 min and centrifuged at 12,000×g rpm for 10 min. Finally, the supernatant was filtered through a 0.22-μm hydrophobic membrane for injection into the LC-MS/MS system.

To accurately determine the amount of CLT in biological samples, we developed a rapid and sensitive LC-MS/MS method. LC-MS/MS analysis was carried out using an Agilent 1200 series RRLC system coupled with an SL autosampler, degasser and SL binary pump as well as an Agilent triple-quadrupole mass spectrometer. CLT and glycyrrhetinic acid IS were separated on a Diamonsil ODS column (50 × 4.6 mm, 1.8 μm) using isocratic elution. The mobile phase consisted of acetonitrile and 0.5% formic acid (80:20, v/v). The total run time was 4.5 min at a flow rate of 0.4 mL per min at 30 °C. Analytes were detected using an electrospray ionization source (ESI) interface in positive ion mode. Multiple reaction monitoring was used to monitor precursor to product ion transitions at m/z 451.3–201.1 (CLT) and m/z 471.3–135.1 (IS). Voltages of fragmentor potential and collision energy were 113 eV and 36 eV for CLT, and 174 eV, and 40 eV for IS. Dwell time was 550 ms. Other instrumental parameters were as follows: gas temperature, 350 °C; gas flow, 8 mL per min; nebulizer pressure, 30 psi; and capillary energy, 4000 V.

**In vivo biodistribution**. Sixty rats were randomly divided into two groups and injected with CLT solution or CLT-AN (1 mg kg⁻¹) via a tail vein. Major tissue samples were collected 0.083, 0.25, 0.5, 1, 2, 4 h after administration. Tissue

samples were rinsed, weighed, and homogenized in two volumes of normal saline (based on tissue weight) using a Precellys 24 lysis instrument (Bertin, France). CLT concentration in tissue homogenates was determined by LC-MS/MS.

Distribution in glomeruli was determined as follows. Kidneys were excised at predetermined times and glomeruli were isolated using a graded sieving technique[60]. In brief, renal cortices were minced and pressed through three successive stainless sieves with pore sizes of 250, 110, and 75 μm using normal saline. Glomeruli were retained on the last sieve, and they were examined on an inverted microscope to ensure high purity with < 5% tubular contamination. Collected glomeruli were centrifuged and the pellet was resuspended in 150 μL of ultrapure water, intermittently ultrasonicated using a probe ultrasonicator at 70 W for 3 min and finally lysed using five freeze-thaw cycles. The resulting glomerular lysate was assayed for total protein using the BCA assay (2 μL; Pierce, USA), and an aliquot (100 μL) was prepared for LC-MS/MS analysis using the liquid-liquid extraction method. Results of glomerular distribution were normalized to the total protein amount.

Pharmacokinetic parameters such as $AUC_{0-t}$, maximal concentration ($C_{max}$), and mean residence time ($MRT_{0-t}$) were calculated using Data and Statistics Software (DAS3.0; Shanghai, China). Relative uptake efficiency ($Re_{glomeruli}$) and concentration efficiency ($Ce_{glomeruli}$) were calculated to evaluate glomeruli targeting. $Re_{glomeruli}$ and $Ce_{glomeruli}$ were calculated using Equation (3) and Equation (4), respectively.

$$Re_{glomeruli} = \left(AUC_{0-t,\ glomeruli}\right)_{CLT-AN} / \left(AUC_{0-t,\ glomeruli}\right)_{CLT} \quad (3)$$

$$Ce_{glomeruli} = \left(C_{max,\ glomeruli}\right)_{CLT-AN} / \left(C_{max,\ glomeruli}\right)_{CLT} \quad (4)$$

**Quantitative RT-PCR.** Total RNA was extracted from isolated renal cortex using the RNAprep Pure Tissue Kit (TianGen, China) following the manufacturer's instructions. First-strand cDNA was obtained by reverse transcription of the total RNA using the TIANScript RT Kit (TianGen, China). The resulting cDNA and corresponding primers were used for SsoFast™ Eva Green Supermix-quantitative real-time PCR to assay levels of MCP-1, ICAM-1, IL-6, IL-1β, PDGF-BB, Col I, Col IV, FN-1, and TGF-β1. The primers were designed based on the mRNA sequences in GenBank and synthesized by Shanghai Shenggong Biotechnology (Shanghai, China). Quantitative QT-PCR was carried out on an iCycler iQ™5 detection system (Bio-Rad, USA) with rat β-actin as internal control. Optimal PCR conditions and primers are summarized in Supplementary Table 3. The normalized fold expressions of tested gene relative to the normal control was calculated based on the $2^{-\Delta\Delta Ct}$ method[61], where Ct is the mean threshold cycle difference.

**In vitro antiproliferative effects of CLT and CLT-AN.** Inhibitory effects of CLT and CLT-AN on PDGF-BB-induced proliferation of HBZY-1 cells were examined using a modified MTT assay. In brief, cells in logarithmic phase were seeded at a density of $5 \times 10^3$ cells/well in 96-well plates. After 24 h, drug-free medium or medium containing PDGF-BB (final concentration, 25 ng mL$^{-1}$), PDGF-BB + CLT or CLT-AN (final concentration, 0.25, 0.5 or 0.75 μg mL$^{-1}$) was added to the corresponding wells and incubated another 24 h. This treatment was followed by addition of 200 μL of MTT solution (0.5 mg mL$^{-1}$ in PBS, pH 7.4) to each well, and cells were incubated in the dark for 4 h at 37 °C. The medium was then removed carefully, and 200 μL of DMSO was added to each well to dissolve the formazan crystals by incubation for 15 min at 37 °C. Optical density (OD) was measured at 570 nm using a Varioskan flash multimode plate reader (Thermo, NH, USA). Replicates of three wells were measured. Untreated cells served as controls and their OD value was defined as 100%.

**Cell cycle arrest.** Cell cycle distributions were assessed by propidium iodide (PI) staining[62]. Exponentially growing HBZY-1 cells ($3 \times 10^5$ per well) were seeded in 6-well plates in triplicate and incubated for 24 h. Drug-free medium or medium with PDGF-BB, PDGF-BB + CLT or CLT-AN was added as described for the modified MTT assay, and the plates were incubated another 24 h. Next, cells were digested, collected, washed, and fixed in 75% ethanol (prechilled to −20 °C) at 4 °C for 30 min. Then the cells were permeabilized with 0.1% Triton X-100 for 15 min and treated with 20 mg mL$^{-1}$ RNaseA for 30 min at 37 °C to digest RNA. PI (20 mg mL$^{-1}$) was added and cells were incubated in the dark for 30 min. Cell cycle distribution was measured using flow cytometer (Cytomics FC 500, Beckman Coulter, USA), and the DNA histogram was analyzed using ModFit LT 2.0 software (USA). Cells exposed to drug-free medium served as the negative control. Flow cytometry gating strategy for cell cycle analysis was shown in Supplementary Fig. 31.

**Cell apoptosis.** The ability of CLT and CLT-AN to induce apoptosis in HBZY-1 cells was measured quantitatively based on FITC-Annexin V and PI double staining[63]. HBZY-1 cells were seeded in the same way as for the cell cycle distribution assay. After 24-h incubation with drug-free medium or medium containing PDGF-BB (final concentration, 25 ng mL$^{-1}$), PDGF-BB + CLT or CLT-AN (final concentration, 0.5 μg mL$^{-1}$), cells were harvested and stained with FITC-Annexin V and PI using a double-staining cell apoptosis analysis kit (BD,

USA) following the manufacturer's instructions. Stained cells were collected to quantify proportions of apoptotic cells using flow cytometry, and the gating strategy for cell apoptosis analysis was shown in Supplementary Fig. 32.

**Cellular uptake.** To explore the mechanism for increased anti-proliferative effect on mesangial cells of CLT-AN, a cellular uptake assay was conducted. In assays to examine dose dependence of uptake, cells were exposed for 4 h to medium containing PDGF-BB + CLT or CLT-AN (final concentrations, 0.5 or 0.75 μg mL$^{-1}$). In assays to examine time dependence of uptake, cells were exposed to PDGF-BB + CLT or CLT-AN at a final concentration of 0.5 μg mL$^{-1}$ for 1, 2 or 4 h. After incubation, cells were rinsed with ice-cold PBS, trypsinized, and centrifuged at 2000×g rpm for 5 min. Pellets were resuspended in 300 μL of ultrapure water and lysed using five freeze-thaw cycles to release CLT. An aliquot (20 μL) of cell lysate was used for total protein determination using the BCA assay reagent kit (Pierce, USA). Another aliquot (100 μL) of cell lysate was prepared using liquid-liquid extraction for LC-MS/MS analysis. Cell uptake measurements were normalized to total protein amount.

**Safety evaluation.** To evaluate the toxicity of free CLT, rats were randomized into three groups: the normal group received PBS; the LD-CLT group treated with 1 mg kg$^{-1}$ CLT and the HD-CLT group treated with 3 mg kg$^{-1}$ CLT. PBS or drugs were administered to rats via tail vein injection once every day for a total of five injections. On day 5, five rats in each group were sacrificed, and the heart, liver, spleen, lung, kidney, and brain were excised immediately. Tissues were washed, fixed, and stained with hematoxylin and eosin (H&E), histological lesions were observed and photographed under a light microscope (Olympus BX53, Tokyo, Japan). All histological analyses were performed by one certified pathologists blinded to sample identity. Before animals were sacrificed, blood (0.3 mL) was collected into non-treated tubes and centrifuged at 5000×g rpm for 10 min. The resulting serum was assayed on a Hitachi 7020 automatic biochemical analyzer (Hitachi, Japan) for the following analytes: BUN, CREA, CK, LDH, ALT, aspartate transaminase (AST), and TBiL. INR of prothrombin time was determined on an ACL TOP 700 coagulation analyzer (Instrumentation Laboratory, Bedford, MA, USA) by the HemosIL reagents according to the manufacturer's protocol.

To evaluate the toxicity of CLT-AN, rats were randomly divided into three groups: the control group received PBS, while the other groups received either free CLT or CLT-AN via tail vein injection at an equivalent dose of 1 mg kg$^{-1}$ once every other day. On days 1, 5, 14, and 21, five rats in each group were sacrificed for the evaluation of histopathology and biochemical parameters as described in toxicity studies of free CLT. Also, the systemic toxicity of empty-AN was examined based on the evaluation of histopathology and biochemical parameters on day 14 or 21 post administration.

**Hepatic distribution of CLT-AN.** To examine the hepatic distribution profile of CLT-AN, liver tissues were isolated from rats at 5 min after FITC-CLT-AN injection ($n = 3$). Isolated liver tissues were fixed with 10% formalin and further dehydrated using 15% (w/v) sucrose, followed by 30% (w/v) sucrose. Tissues were snap-frozen in OCT compound, and 4-μm frozen sections were prepared and stained with mouse anti-CD68 antibody (1:200; Abcam, ab31630), followed by Alexa Fluor®594 goat anti-mouse antibody (1:100; Abcam, ab150116). For nuclear staining, DAPI (4′, 6-diamidino-2-phenylindole) was used. Representative photomicrographs were taken by CLSM.

**Statistical analysis.** All data are reported as mean ± standard deviation (s.d.). Statistical analysis was performed by SPSS 20.0 software (USA). Inter-group differences were analyzed using the Student's t-test when two groups were compared, or one-way analysis of variance (ANOVA) with Tukey post hoc test when multiple groups were compared. A P-value < 0.05 was considered statistically different.

**Data availability.** The data that support the findings of this study are available from the corresponding author upon reasonable request.

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

## Acknowledgements

This work was supported by the National Natural Science Foundation of China (81690261). We thank Dr Jun Gao (Department of Toxicological Inspection, Sichuan Center for Disease Prevention and Control, Chengdu, China) for the support of histological analysis.

## Author contributions

Z.Z. conceived and planned the study. L.G. carried out the experiments, generated and analyzed data, and wrote the original manuscript. S.L., Z.D., M.Z., and P.L. helped with animal and cell studies. Y.F., X.S., Y.H., and Z.Z. helped with manuscript editing.

## Additional information

**Competing interests:** The authors declare no competing financial interests.

