## [Peer Review file · Nature Communications]

Reviewers' comments:

Reviewer #1 (Remarks to the Author):

This manuscript presented us a promising mesangial cell-targeted drug delivery system for mesangial cell-mediated glomerulonephritis therapy, using an immunosuppressive, anti-inflammatory and anticancer extraction agent celastrol (CTL) from the traditional Chinese medicine-Thunder of God Vine. A complete story was told by the authors, showing us why they chose CTL as the therapeutic agent, why they decided to prepare a drug delivery system and how this mesangial cell-targeted drug delivery system performed good efficiency in attenuating proteinuria, inflammation, glomerular hypercellularity and ECM deposition in disease model rats with reduced systemic toxicity.

From my point of view, this manuscript was designed with good logic and completed with abundant experiments, which is well worth appreciation. However, there are still some points that need attention.

1. Since free CTL showed a superior therapeutic efficacy than MPA in Fig. 1, authors should provide data of distribution and systemic toxicity of free CTL at the very beginning as they conducted in Fig. 8 and Fig. 9 to demonstrate the necessity for preparing a drug delivery system.
2. The authors should tell us why they chose albumin nanoparticles as drug carrier rather than something else. There are many nano delivery systems available.
3. After encapsulating CTL, the average size of nanoparticles was still around 95 nm, almost the same as before. What's the mechanism of albumin nanoparticles encapsulating CTL and why there was no size difference?
4. The drug loading efficiency of the albumin nanoparticles is only 2.35%. So, the mesangial cells are essentially "eating" a lot of albumin. All experiments should include an empty nanoparticle control.
5. In Fig. 8 and Fig. 9, nanoparticles were largely distributed in liver even more than kidney, but the inherent hepatotoxicity of CTL was weak. Authors claimed that most nanoparticles were accumulated in Kupffer cells. Evidence, for example, co-localization of nanoparticles and Kupffer cells rather than hepatocytes should be provided.
6. In the first paragraph, authors said that "Pharmacological treatments against inflammation and glomerular disorders may slow GN progression and related mortality. However, no effective treatment is currently available for stopping or reversing the disease progression." Based on this, I think survival experiment should be added in the manuscript.

Reviewer #2 (Remarks to the Author):

Guo et al report on a role of celastrol (CTL), a compound derived from a traditional Chinese medicine. The reports is in three parts: 1) Celastrol is effective in experimental mesangial proliferative glomerulonephritis 2) combined with 95nm albumin nanoparticles (AN) it is relatively selective for the mesangium and more effective 3) there is less (acute) toxicity when CTL is used in AN.

Celestrol inhibits NF- κ B and has been used and published in a number of animal models of inflammatory diseases, including in the kidney, diabetic nephropathy and acute kidney injury. Nanoparticles have previously been designed to localise to mesangial areas (cited in the current

report). This paper combines the elements of both these types of studies, that is celastrol as an anti-inflammatory agent, and targeting mesangial cells to essentially show that it can attenuate (treatment started at onset of disease) a number of cardinal manifestations of experimental mesangial proliferative glomerulonephritis. The authors also perform limited but interesting toxicity studies that suggest that CTL administered in these nanoparticles limits CTL toxicity (despite accumulation in the liver).

1. The Thy 1 model used is a valid and well characterized model of glomerulonephritis. However, there are several caveats: mesangiolytic is critical to the model, but not a feature of IgA nephropathy, the disease it models (a significant omission in the paper is the lack of mention of IgA as the key human disease the authors plan to treat). The role of proteinuria in this model is a little uncertain, as it occurs in this context – and in that of repair. A chronic two-hit model (eg nephrectomy + OX-7) has been described in this model and it would be reassuring to see an intervention study after disease has been established in this sort of model (or even in the single shot model).

2. Treatments were given after binding of OX-7 but not after disease is established, limiting the capacity to draw conclusions about treatment of disease (which, as opposed to elucidating pathogenesis, seems to be the major aim of the studies).

3. With regard to the studies, most seem to have been well performed. Provision of negative controls for all immunohistochemistry would be relevant to show specificity of staining signal.

4. Unfortunately interpretation of the results presented in Figure 5, 6 e and f, and 7 is not possible (though the text does interpret them as showing beneficial reductions) as no statistical significance is claimed or shown. This is a significant drawback.

It is surprising not to see some sort of direct assessment in vivo and in vitro of NF- κ B in this study given celestrol's potential role as an NF- κ B inhibitor.

5. It is important not to overplay the lack of toxicity of the drug-AN complex given the limited nature of the studies, the accumulation of the drug in the liver, and the arrows in Supp 8 in CLT-AN treated animal showing histological evidence of toxicity.

6. There are a number of statements that are either inaccurate or seem to overstate the significance of the findings as they stand currently:

a) It is not accurate that there is no effective treatment available for stopping disease processes in glomerulonephritis (introduction, page 3 lines 39-40). It is also worth noting that MMF, the comparator in the authors' studies, is not a proven (or commonly used) therapy for IgA (mesangial proliferative) nephropathy, and the authors should be careful not to imply that it is such.

b) Please cite compelling references as to the proven efficacy of TGV in glomerulonephritis or modify omit statements on page 3 lines 45-46, and lines 52-53; Page 7 lines 133-134.

c) The significance of individual results is overstated on several occasions: for example the inference that this treatment has direct clinical relevance, that the reduction in acute proteinuria "may have profound clinical significance" in that it translates to chronic disease.

d) I am not sure that the interesting studies that they have performed on toxicity justify conclusions such as "CLT-AN represents a safe and effective treatment option for MsPGN and relevant glomerular diseases" as stated in the abstract.

7. Stylistically qualifying adjectives and adverbs that add little to the results are used: for examples "dramatic" and "dramatically" that are used on multiple occasions. Could the manuscript be modified to allow the results to stand for themselves without unnecessary commentary and qualifications.

8. Minor comments: There are grammatical and spelling errors that should be corrected. The yellow type on a white background label in FIG 2e is not readable. Figure 10 is redundant (unless

it is journal style to encourage the publication of summary cartoons).

Reviewer #3 (Remarks to the Author):

This interesting study evaluates the effect of celastrol in a mesangial proliferative glomerulonephritis rat model showing that its administration in a developed mesangial cells-targeted drug delivery system via albumin nanoparticles keeps the efficacy dramatically decreasing systemic toxicity. The results regarding liver safety of the albumin nanoparticles are quite consistent; however, some aspects of liver safety assessment need needs further clarification.

1. The authors should comment the background data on hepatotoxicity of celastrol
2. The authors selected AST and ALT as "liver function tests" (line 359), which is not correct as aminotransferases are non-specific markers of liver injury but they do not measure liver "function" (Senior Clin Pharmacol Ther 2012). Liver function is better assessed by bilirubin and international normalized ratio. Why did not they assess these parameters?
3. The authors shown a very good liver safety (as measured by the level of aminotransferases) of celastrol in acute studies (days 1 and 5) but chronic toxicity studies would be also required in preclinical studies.

Responses to the reviewers' comments:

Reviewer #1 (Remarks to the Author):

This manuscript presented us a promising mesangial cell-targeted drug delivery system for mesangial cell-mediated glomerulonephritis therapy, using an immunosuppressive, anti-inflammatory and anticancer extraction agent celastrol (CLT) from the traditional Chinese medicine-Thunder of God Vine. A complete story was told by the authors, showing us why they chose CLT as the therapeutic agent, why they decided to prepare a drug delivery system and how this mesangial cell-targeted drug delivery system performed good efficiency in attenuating proteinuria, inflammation, glomerular hypercellularity and ECM deposition in disease model rats with reduced systemic toxicity. From my point of view, this manuscript was designed with good logic and completed with abundant experiments, which is well worth appreciation. However, there are still some points that need attention.

1. Since free CTL showed a superior therapeutic efficacy than MPA in Fig. 1, authors should provide data of distribution and systemic toxicity of free CTL at the very beginning as they conducted in Fig. 8 and Fig. 9 to demonstrate the necessity for preparing a drug delivery system.

Response: Thanks for the reviewer's constructive comment. Per your suggestion, we carried out the biodistribution and systemic toxicity study of free CLT. Results and experiments were provided in the new section "**CLT caused severe systemic toxicity due to its off-target distribution**" and **Method section** in the revised manuscript. Also, representative photomicrographs of main tissue sections stained with hematoxylin & eosin (H&E) and the main serum biological parameters after free CLT treatment were provided in **Fig. 2a** and **Supplementary Fig. 7**. C_{max} and AUC_{0-4h} of CLT in main organs were shown in **Fig. 2b,c**. These data confirmed that free CLT induced severe cardiac toxicity, hepatotoxicity or neurotoxicity due to its off-target distribution, which drove us to develop a targeted drug delivery system to improve the safety of CLT without compromising its efficacy to treat glomerulonephritis.

2. The authors should tell us why they chose albumin nanoparticles as drug carrier rather than something else. There are many nano delivery systems available.

Response: Regarding selection of appropriate nanocarriers, human serum albumin (HSA)-based nanoparticles show several advantages:

1) Compared with polymeric nanoparticles and polymeric micelles, albumin nanoparticles (ANs) prepared from HSA are biodegradable, non-toxic and non-immunogenic, which represents a highly safe and translatable delivery system. For example, the HSA-bound paclitaxel nanoparticles (Abraxane) have been approved for treating metastatic breast cancers by the US Food and Drug Administration in 2005¹.

2) Compared with liposomes or solid lipid nanoparticles, HSA-based nanoparticles can easily be prepared with sufficient stability. In our preliminary study, we prepared ANs using a facile and highly repeatable ultrasonication method (detailed protocol was shown in the **Methods section**). ANs with varying sizes were obtained by adjusting the amount of soybean oil and ultrasonication power. However, liposomes or solid lipid nanoparticles usually prepared using thin film hydration method which is difficult to control size distribution and stability. Also, stabilizers such as surfactants were added or the thin film hydration method was used in combination with ultrasonication to prepare stable and uniform nanoparticles, which is very time-consuming and expensive.

3) Given the high binding affinity of CLT to albumin (the human plasma protein binding rate of CLT > 80%)^{2,3}, incorporating CLT into HSA nanoparticles was highly feasible.

Thus, we chose albumin nanoparticles as the nanocarrier. A statement on the advantages of albumin nanoparticles was provided and highlighted in the **Discussion section** (page 23 lines 490-494).

Reference:

- 1 Hawkins, M. J., Soon-Shiong, P. & Desai, N. Protein nanoparticles as drug carriers in clinical medicine. *Advanced drug delivery reviews* **60**, 876-885 (2008).
- 2 Yuan, L., Chen, Y., Zhou, L., Zheng, L. & Cao, W. Determination of protein binding rates of celastrol in different species of plasma by HPLC. *Chinese Journal of Hospital Pharmacy* **33**, 175-179 (2013).
- 3 Zhang, T. *et al.* Spectroscopic study of binding reaction between celastrol and bovine serum albumin. *Journal of Food Science* **30**, 130-133 (2009).

3. After encapsulating CTL, the average size of nanoparticles was still around 95 nm, almost the same as before. What's the mechanism of albumin nanoparticles encapsulating CTL and why there was no size difference?

Response: Thanks for the reviewer's comment. As demonstrated in the Section "ANs size-dependently accumulated at glomerular mesangial cells", albumin nanoparticles (ANs) with an average diameter of ~95 nm (AN-95) showed maximal mesangial cellular localization than the other two formulations (AN-75 or AN-130), which indicate that ~95 nm represents the optimal size range that maximizes mesangial cell targeting of ANs. Thus, to selectively deliver as much CLT as possible to mesangial cells, we encapsulated CLT into ANs and attempted to keep the particle size of CLT-loaded albumin nanoparticles (CLT-AN) approximately 95 nm in the following studies.

To prepare blank AN-95, we used following conditions: briefly, **25 mg of soybean oil** were dissolved in 2 mL of a mixture solvent of methylene chloride and ethyl acetate to form the organic phase, and 1 mL of 20% (w/v) HSA dispersed in 10 mL distilled water were added to the organic phase. Then, the dispersion was intermittently ultrasonicated at **330 W for 8 min** using a probe ultrasonicator (Xinzhi Biotechnology, Ningbo, China). The organic solvent was subsequently removed by vacuum rotary evaporation at 37 °C,

following which the blank AN-95 was obtained. As measured by dynamic light scattering, its average hydrodynamic diameter was 94.61 ± 2.22 nm.

In our preliminary study, we attempted to prepare CLT-AN by directly introducing predetermined amounts of CLT in preparation protocol of blank AN-95 above, however, CLT-AN obtained was approximately 150 nm rather than 95 nm in diameter. Thus, to obtain CLT-AN with an average diameter of ~95 nm, **we optimized the formulation and preparation of CLT-AN as follows:** in brief, **45 mg of soybean oil and 7 mg of CLT** were dissolved in 2 mL of a mixture solvent of methylene chloride and ethyl acetate to form the organic phase, and 1 mL of 20% (w/v) HSA dispersed in 10 mL distilled water were added to the organic phase. Then, the dispersion was intermittently ultrasonicated at **490 W for 8 min** using a probe ultrasonicator. The organic solvent was subsequently removed by vacuum rotary evaporation, following which CLT-AN was obtained and its average hydrodynamic diameter was 95.97 ± 0.22 nm as expected.

The mechanism of albumin nanoparticles encapsulating CTL may involve both emulsification and chemical cross-linking of albumin: using ultrasonication, the emulsification will occur in the mixture of aqueous HSA solution and water-insoluble organic phase containing CLT and soybean oil, and an oil-in-water emulsion will thus be formed⁴⁻⁶. The superoxide (HO_2) generated sonochemically from water and oxygen by acoustic cavitation is a protein cross-linking agent^{6,7}, which further cross-linked HSA together by disulphide bond between cysteine residues and formed the protein shell around the inner non-aqueous droplet containing CLT^{4,8}. Solvent-free CLT-AN is then obtained after evaporation of the organic solvent^{9,10}.

Reference:

- 4 Han, Y., Radziuk, D., Shchukin, D. & Moehwald, H. Stability and size dependence of protein microspheres prepared by ultrasonication. *Journal of Materials Chemistry* **18**, 5162-5166 (2008).
- 5 Suslick, K. S. *Ultrasound: its chemical, physical, and biological effects*. (VCH Publishers, 1988).
- 6 Suslick, K. S. & Grinstaff, M. W. Protein microencapsulation of nonaqueous liquids. *Journal of the American Chemical Society* **112**, 7807-7809 (1990).
- 7 Suslick, K., Grinstaff, M., Kolbeck, K. & Wong, M. Characterization of sonochemically prepared proteinaceous microspheres. *Ultrasonics Sonochemistry* **1**, S65-S68 (1994).
- 8 Avivi, Nitzan, Y., Dror, R. & Gedanken, A. An easy sonochemical route for the encapsulation of tetracycline in bovine serum albumin microspheres. *Journal of the American Chemical Society* **125**, 15712-15713 (2003).
- 9 Desgouilles, S. *et al.* The design of nanoparticles obtained by solvent evaporation: a comprehensive study. *Langmuir* **19**, 9504-9510 (2003).
- 10 Reis, C. P., Neufeld, R. J., Ribeiro, A. J. & Veiga, F. Nanoencapsulation I. Methods for preparation of drug-loaded polymeric nanoparticles. *Nanomedicine*:

4. The drug loading efficiency of the albumin nanoparticles is only 2.35%. So, the mesangial cells are essentially “eating” a lot of albumin. All experiments should include an empty nanoparticle control.

Response: The authors agreed that an empty nanoparticle control should be provided in each set of study. However, given the limited time available and high cost of animal studies, we cannot accomplish all experiments again. However, we have strictly added the empty nanoparticle control in our later supplementary experiments including the intervention study of CLT or CLT-AN after disease model was established and chronic toxicity assays of CLT or CLT-AN (**Supplementary Figs. 15 and 21**). As shown in **Supplementary Fig. 15**, the exposure of anti-thy1.1 nephritic rats to empty albumin nanoparticles (Empty-AN) did not significantly alter proteinuria, inflammation, glomerular hypercellularity and extracellular matrix expansion, suggesting that the enhanced therapeutic efficacy of CLT-AN was mainly attributed to the targeted delivery of CLT to the mesangial cells but not the synergistic effects of CLT and Empty-AN. Also, as demonstrated in **Supplementary Fig. 21**, long time exposure of Empty-AN in normal rats did not induce any visible signs of systemic toxicity.

5. In Fig. 8 and Fig. 9, nanoparticles were largely distributed in liver even more than kidney, but the inherent hepatotoxicity of CTL was weak. Authors claimed that most nanoparticles were accumulated in Kupffer cells. Evidence, for example, co-localization of nanoparticles and Kupffer cells rather than hepatocytes should be provided.

Response: Thanks for the reviewer’s constructive comment. Per your suggestion, we have investigated the distribution profile of CLT-AN in the liver. These results and experiments were supplemented in “**CLT-AN altered the tissue biodistribution profiles of CLT in rats**” and **Methods** in the revised manuscript. As shown in **Supplementary Fig. 22**, immunofluorescent analysis with a macrophage marker CD68 revealed that CLT-AN were largely retained in CD68 positive Kupffer cells rather than in hepatocytes, which might further explain the reduced hepatic toxicity of CLT-AN.

6. In the first paragraph, authors said that “Pharmacological treatments against inflammation and glomerular disorders may slow GN progression and related mortality. However, no effective treatment is currently available for stopping or reversing the disease progression.” Based on this, I think survival experiment should be added in the manuscript.

Response: Thanks for the reviewer’s comment. In our study, we have examined the therapeutic efficacy and potential mechanisms of CLT or CLT-AN in anti-Thy1.1 nephritic rats, a well-established animal model for mesangial proliferative glomerulonephritis (MsPGN). Animal death was not observed in all treatment groups

during the course of these experiments. Thus, we did not perform the survival experiment.

Reviewer #2 (Remarks to the Author):

Guo et al report on a role of celastrol (CTL), a compound derived from a traditional Chinese medicine. The reports is in three parts: 1) Celastrol is effective in experimental mesangial proliferative glomerulonephritis 2) combined with 95nm albumin nanoparticles (AN) it is relatively selective for the mesangium and more effective 3) there is less (acute) toxicity when CTL is used in AN.

Celestrol inhibits NF-kB and has been used and published in a number of animal models of inflammatory diseases, including in the kidney, diabetic nephropathy and acute kidney injury. Nanoparticles have previously been designed to localise to mesangial areas (cited in the current report). This paper combines the elements of both these types of studies, that is celastrol as an anti-inflammatory agent, and targeting mesangial cells to essentially show that it can attenuate (treatment started at onset of disease) a number of cardinal manifestations of experimental mesangial proliferative glomerulonephritis. The authors also perform limited but interesting toxicity studies that suggest that CTL administered in these nanoparticles limits CTL toxicity (despite accumulation in the liver).

1. The Thy 1 model used is a valid and well characterized model of glomerulonephritis. However, there are several caveats: mesangiolysis is critical to the model, but not a feature of IgA nephropathy, the disease it models (a significant omission in the paper is the lack of mention of IgA as the key human disease the authors plan to treat). The role of proteinuria in this model is a little uncertain, as it occurs in this context – and in that of repair. A chronic two-hit model (*eg* nephrectomy + OX-7) has been described in this model and it would be reassuring to see an intervention study after disease has been established in this sort of model (or even in the single shot model).

Response: The authors would like to thank the reviewer for the constructive comments. Mesangial hypercellularity and extracellular matrix (ECM) expansion are key pathological features of human mesangial proliferative glomerulonephritis (MsPGN) including IgA nephropathy and non-IgA nephropathy^{11,12}. These glomerular lesions are also seen in many secondary glomerular diseases such as lupus GN and diabetic nephropathy¹³. Thus, the search for agents favorably regulating MCs proliferation and deposition of ECM proteins is of substantial clinical importance. The rat anti-Thy1.1 model provides a relatively rapid and reliable model of MsPGN¹⁴, which is extensively used for testing novel pharmacological compounds that interfere with mesangial cell proliferation as well as matrix overproduction^{11,15-20}. This model can be quickly induced in rats by either polyclonal or monoclonal antibodies against Thy1.1 antigen present on the surface of MCs, and it clearly encompasses important processes of MsPGN including infiltration of circulating inflammatory cells, marked proliferation of MCs and accumulation of ECM. Thereby, this experimental animal model was chosen to evaluate the therapeutic effects of CLT and CLT-AN on MsPGN in our study.

1) To analyze the therapeutic effects of CLT or CLT-AN, we investigated their effects against proteinuria, inflammation, glomerular hypercellularity and ECM deposition in anti-Thy1.1 nephritic rats. We did not mention their anti-immunoglobulin A (IgA) effects mainly due to following reasons. **Complement-dependent mesangiolysis mediates the initiation of anti-thy1.1 model, which is not a feature of IgA nephropathy. IgA nephropathy is characterized by mesangial IgA deposition²¹, which is not essentially present in anti-thy1.1 model²². Although both anti-thy1.1 nephritis and IgA nephropathy share similar features of increased MCs- proliferation and mesangial matrix-expansion, this model cannot be regarded as an appropriate IgA nephropathy model²². Thus, we did not examine the effects of CLT or CLT-AN on IgA deposition in the anti-Thy1.1 rat model.** In our study, attentions have been focused on developing a mesangial cells-targeted drug delivery system via albumin nanoparticles (ANs) to improve the safety and efficacy of CLT. CLT-AN has been proven with improved anti-inflammation, anti-proliferation and anti-fibrosis effects while significantly reduced toxicity compared to free CLT. As a result, CLT-AN represents a viable and efficient strategy to attenuate mesangial hypercellularity and ECM expansion in MCs-mediated glomerulonephritis including IgA nephropathy, lupus GN and diabetic nephropathy. However, the detailed molecular mechanisms underlying their function remained to be explored. IgA nephropathy has been the most common primary glomerulonephritis worldwide, and a systematic investigation of the effects of CLT-AN on the glomerular IgA deposition or other lesions in IgA nephropathy has been ongoing.

2) The authors agree with the reviewer that proteinuria occurs in both the disease development and repair stage of anti-Thy1.1 nephritis model^{18,23}. Proteinuria denotes a sign of glomerular diseases and represents a marker of injury to the glomerular permeability barrier^{24,25}. Reduction of proteinuria is often associated with beneficial effect of treatment^{11,15,16,18-20,26,27}. In our study, to evaluate the therapeutic efficacy of CLT or CLT-AN against anti-Thy1.1 nephritis, 24-h urine samples were collected from all rats and urinary protein concentration was determined on day 5 after disease induction. Experimental groups included normal control, anti-Thy1.1 nephritis control, and drug-treated groups. As show in **Figs. 1a, 5a,e and Supplementary Fig. 4a**, CLT or CLT-AN treatment led to significantly lower urinary protein excretion compared to the anti-Thy1.1 nephritis control, suggesting the excellent benefits of CLT or CLT-AN against proteinuria in anti-Thy1.1 nephritis. However, the detailed molecular mechanisms underlying their function remained to be explored in our future study.

3) In some studies, the combination of OX7 injection and uninephrectomy or repeated injections of OX7 have been used to produce a progressive MsPGN model^{28,29}. However, given the limited time and financial support, we only performed the intervention study of free CLT or CLT-AN after the single shot disease model was established, and have added the corresponding experiments and results in the **Section “CLT attenuated proteinuria and glomerular lesions in anti-Thy1.1 nephritis rats”** and **Section “CLT-AN enhanced therapeutic efficacy of CLT in anti-Thy1.1 nephritic rats”** and **Methods**

section accordingly in the revised manuscript. **Please see responses to Question 2 for an in-depth discussion on the topic.**

Reference:

- 11 Floege, J., Eng, E., Young, B. A., Couser, W. G. & Johnson, R. J. Heparin suppresses mesangial cell proliferation and matrix expansion in experimental mesangioproliferative glomerulonephritis. *Kidney international* **43**, 369-380 (1993).
- 12 Floege, J. *et al.* Increased synthesis of extracellular matrix in mesangial proliferative nephritis. *Kidney international* **40**, 477-488 (1991).
- 13 Scindia, Y. M., Deshmukh, U. S. & Bagavant, H. Mesangial pathology in glomerular disease: targets for therapeutic intervention. *Advanced drug delivery reviews* **62**, 1337-1343 (2010).
- 14 Jefferson, J. A. & Johnson, R. J. Experimental mesangial proliferative glomerulonephritis (the anti-Thy-1.1 model). *Journal of nephrology* **12**, 297-307 (1998).
- 15 Chen, Y.-M. *et al.* Pentoxifylline attenuates experimental mesangial proliferative glomerulonephritis. *Kidney international* **56**, 932-943 (1999).
- 16 Chiang, C., Sheu, M., Hung, K., Wu, K. & Liu, S. Honokiol, a small molecular weight natural product, alleviates experimental mesangial proliferative glomerulonephritis. *Kidney international* **70**, 682-689 (2006).
- 17 Daniel, C., Ziswiler, R., Frey, B., Pfister, M. & Marti, H.-P. Proinflammatory effects in experimental mesangial proliferative glomerulonephritis of the immunosuppressive agent SDZ RAD, a rapamycin derivative. *Nephron Experimental Nephrology* **8**, 52-62 (2000).
- 18 Suana, A. *et al.* Single application of low-dose mycophenolate mofetil-OX7-immunoliposomes ameliorates experimental mesangial proliferative glomerulonephritis. *Journal of Pharmacology and Experimental Therapeutics* **337**, 411-422 (2011).
- 19 Wan, Y. *et al.* Multi-glycoside of *Tripterygium wilfordii* Hook f. ameliorates proteinuria and acute mesangial injury induced by anti-Thy1. 1 monoclonal antibody. *Nephron Experimental nephrology* **99**, e121-e129 (2005).
- 20 Ziswiler, R., Steinmann-Niggli, K., Kappeler, A., Daniel, C. & Marti, H.-P. Mycophenolic acid: a new approach to the therapy of experimental mesangial proliferative glomerulonephritis. *Journal of the American Society of Nephrology* **9**, 2055-2066 (1998).
- 21 Mestecky, J. *et al.* IgA nephropathy: molecular mechanisms of the disease. *Annual Review of Pathology: Mechanisms of Disease* **8**, 217-240 (2013).
- 22 Eitner, F., Boor, P. & Floege, J. Models of IgA nephropathy. *Drug Discovery Today: Disease Models* **7**, 21-26 (2010).
- 23 Reinhart, G. A. & Cox, B. F. Models of Renal Insufficiency: The Anti-Thy-1.1

- Model of Acute Proliferative Glomerulonephritis. *Current Protocols in Pharmacology*, 5.21. 21-25.21. 13 (2001).
- 24 Keane, W. F. Proteinuria: its clinical importance and role in progressive renal disease. *American journal of kidney diseases* **35**, S97-S105 (2000).
- 25 Schaefer, L. *et al.* Nephric expression is increased in anti-Thy1. 1-induced glomerulonephritis in rats. *Biochemical and biophysical research communications* **324**, 247-254 (2004).
- 26 Chen, Y., Lin, S., Chiang, W., Wu, K. & Tsai, T. Pentoxifylline ameliorates proteinuria through suppression of renal monocyte chemoattractant protein-1 in patients with proteinuric primary glomerular diseases. *Kidney international* **69**, 1410-1415 (2006).
- 27 Iseki, K., Ikemiya, Y., Iseki, C. & Takishita, S. Proteinuria and the risk of developing end-stage renal disease. *Kidney international* **63**, 1468-1474 (2003).
- 28 Krämer, S. *et al.* Mycophenolate mofetil slows progression in anti-thy1-induced chronic renal fibrosis but is not additive to a high dose of enalapril. *American Journal of Physiology-Renal Physiology* **289**, F359-F368 (2005).
- 29 Morita, H. *et al.* Induction of irreversible glomerulosclerosis in the rat by repeated injections of a monoclonal anti-Thy-1.1 antibody. *Nephron* **60**, 92-99 (1992).

2. Treatments were given after binding of OX-7 but not after disease is established, limiting the capacity to draw conclusions about treatment of disease (which, as opposed to elucidating pathogenesis, seems to be the major aim of the studies).

Response: Thanks for the reviewer's constructive comment. The authors agreed with the reviewer that it would be inappropriate to draw such a conclusion about the capacity of CLT or CLT-AN in the treatment of disease based on the animal model and studies reported in the previous manuscript. Per your suggestion, we have performed the intervention study of free CLT or CLT-AN after anti-Thy1.1 nephritis was established in the revised manuscript (**Supplementary Figs. 4-6, Fig. 5, Supplementary Figs. 12 and 14**). According to our preliminary experiment (**Supplementary Figs. 3**), marked proteinuria, glomerular infiltration of circulating monocytes, glomerular hypercellularity and ECM accumulation, could be observed in anti-Thy1.1 nephritic rats at day 2 after disease induction compared to normal control. Thus, we sought to understand whether CLT or CLT-AN treatment initiated at day 2 after disease induction can attenuate proteinuria and glomerular lesions in anti-Thy1.1 nephritis rats. For the convenience of understanding the intervention study setup, we included a schematic of induction of anti-Thy1.1 nephritis and intravenous treatments in **Supplementary Fig. 23**. We also added the corresponding experiments and results in **Section "CLT attenuated proteinuria and glomerular lesions in anti-Thy1.1 nephritis rats"** and **Section "CLT-AN enhanced therapeutic efficacy of CLT in anti-Thy1.1 nephritic rats"** and **Methods section** accordingly.

For the intervention study of free CLT, early CLT treatment significantly attenuated

proteinuria, inflammation, glomerular hypercellularity and ECM deposition in anti-Thy1.1 nephritic rats (**Fig. 1; Supplementary Figs. 1,2**). Furthermore, the protective effect of CLT was well demonstrated in the late intervention study when treatment was initiated after the nephritis was established (**Supplementary Figs. 4-6**). For the intervention study of CLT-AN, compared to free CLT, CLT-AN showed better therapeutic efficacy regardless of when it was initiated (**Fig. 5, Supplementary Figs. 11-14**). Together, these results suggest that CLT represents an effective agent in the treatment of MsPGN, and the enhanced therapeutic efficacy could be achieved by encapsulating CLT into albumin nanoparticles (ANs).

3. With regard to the studies, most seem to have been well performed. Provision of negative controls for all immunohistochemistry would be relevant to show specificity of staining signal.

Response: Thanks for the reviewer's comment. The immunohistochemical staining in our study was performed as previously reported³⁰. We replaced the primary antibody with buffer in negative controls^{18,20}. The representative photomicrographs of immunostaining for ED-1, MCP-1, ICAM-1, IL-6, IL-1 β , NF- κ B, α -SMA, PCNA, PDGF-BB, Col I, Col IV, FN-1, TGF- β ₁ in kidney tissue sections taken from anti-Thy1.1 nephritic rats with the corresponding primary antibody or PBS were provided in **Supplementary Fig. 24**. As it demonstrated, no positive staining was observed in all negative controls, suggesting the staining signal was produced by the specific binding between the primary antibody and antigen.

Reference:

- 30 Wang, X. et al. Mechanistic Studies of a Novel Mycophenolic Acid–Glucosamine Conjugate That Attenuates Renal Ischemia/Reperfusion Injury in Rat. *Molecular pharmaceutics* 11, 3503-3514 (2014).
- 18 Suana, A. et al. Single application of low-dose mycophenolate mofetil-OX7-immunoliposomes ameliorates experimental mesangial proliferative glomerulonephritis. *Journal of Pharmacology and Experimental Therapeutics* **337**, 411-422 (2011).
- 20 Ziswiler, R., Steinmann-Niggli, K., Kappeler, A., Daniel, C. & Marti, H.-P. Mycophenolic acid: a new approach to the therapy of experimental mesangial proliferative glomerulonephritis. *Journal of the American Society of Nephrology* **9**, 2055-2066 (1998).

4. Unfortunately interpretation of the results presented in Figure 5, 6 e and f, and 7 is not possible (though the text does interpret them as showing beneficial reductions) as no statistical significance is claimed or shown. This is a significant drawback. It is surprising not to see some sort of direct assessment in vivo and in vitro of NF- κ B in this study given celestrol's potential role as an NF- κ B inhibitor.

Response: Thank you very much for pointing out the omissions in our manuscript.

1) For the results of quantitative RT-PCR presented in Figure 5, 6 e and f, and 7, we have re-analyzed all the quantitative gene expression data and normalized fold expressions relative to the normal control based on $2^{-\Delta\Delta CT}$ method³¹, and the corresponding results were displayed in **Supplementary Table 4**. We also reformatted and added the statistical analysis in **Figs. 6a-d, 7e and 8a-d** in the revised manuscript. As demonstrated, all the up-regulated mRNA expression of MCP-1, ICAM-1, IL-6, IL-1 β , PDGF-BB, Col I, Col IV, FN-1 and TGF- β ₁ in anti-thy1.1 nephritic rats could be significantly down-regulated by CLT or CLT-AN, and CLT-AN showed a down-regulatory effect to a greater extent than free CLT.

2) Per your suggestion, we supplemented the immunohistochemical analysis to evaluate the effects of CLT and CLT-AN on the expression of NF- κ B in anti-Thy1.1 nephritic rats on day 1 after disease induction. As shown in **Supplementary Fig. 16**, the increased immunostaining scores for NF- κ B in anti-Thy1.1 nephritic rats were significantly reduced by CLT and CLT-AN, and CLT-AN attenuated the increased scores to a greater extent than free CLT.

Reference:

31 Livak, K. J. &Schmittgen, T. D. Analysis of relative gene expression data using real-time quantitative PCR and the $2^{-\Delta\Delta CT}$ method. *Methods* **25**, 402-408 (2001).

5. It is important not to overlay the lack of toxicity of the drug-AN complex given the limited nature of the studies, the accumulation of the drug in the liver, and the arrows in Supp 8 in CLT-AN treated animal showing histological evidence of toxicity.

Response: Thanks for the reviewer's comment. For the acute toxicity study of free CLT or CLT-AN (**Fig. 9a; Supplementary Fig. 18**), free CLT treatment induced severe atrophy of myocardial cells and myofibrillar loss in the heart, dilatation of blood sinus and loss of hepatic cords in the liver, pyknosis of neuron and red neurons in the brain. **In contrast, CLT-AN showed greatly reduced cardiac toxicity or hepatotoxicity and no signs of neurotoxicity.** We have corrected the corresponding statements and highlighted in red in the revised manuscript (page 19 lines 414-415 and page 20 lines 417-419, 422-423). Besides, we performed a long time toxicity study of CLT or CLT-AN at day 14 or 21 post intravenous administration based on evaluation of biochemical parameters and histopathology. As shown in **Supplementary Figs. 19 and 20**, long time CLT-AN treatment consistently induced significantly reduced cardiac toxicity or hepatotoxicity but no neurotoxicity compared to free CLT. These results suggest that encapsulating CLT in albumin nanoparticles is a promising strategy for reducing the systemic toxicity of CLT.

6. There are a number of statements that are either inaccurate or seem to overstate the significance of the findings as they stand currently:

a) It is not accurate that there is no effective treatment available for stopping disease processes in glomerulonephritis (introduction, page 3 lines 39-40). It is also worth noting that MMF, the comparator in the authors' studies, is not a proven (or commonly used) therapy for IgA (mesangial proliferative) nephropathy, and the authors should be careful not to imply that it is such.

Response: The authors would like to thank the reviewer for the thoughtful comments.

1) Per your suggestion, we deleted this sentence "However, no effective treatment is currently available for stopping or reversing the disease progression" in the revised manuscript.

2) The authors agree with the reviewer that MMF is not a proven or commonly used therapy for IgA (mesangial proliferative) nephropathy. **We corrected the statement and highlighted it in red in the revised manuscript (page 3 lines 51-52).** Given the complex pathogenetic mechanisms, no specific or optimal treatment to date is available for patients with IgA nephropathy³². Many studies evaluated the role of immunosuppressive medications or cytotoxic agents in the treatment of IgA nephropathy. As the first-line immunosuppressive treatment of IgA nephropathy, prednisone or prednisolone have shown benefits in some clinic trials³³⁻³⁵, however, no apparent therapeutic efficacy in other studies^{36,37}. Similarly, the benefits of MMF as a novel immunosuppressant for the treatment of IgA nephropathy also remain controversial or uncertain³⁸⁻⁴¹. Some studies suggest the cyclophosphamide may have benefits in IgA nephropathy^{35,42}, but its risk or toxicity remains a big concern⁴³. All these conflicting results or toxicity concerns make it difficult to choose a suitable therapeutic control in our study. Considering the main purpose of our study is to evaluate the effects of CLT on proteinuria, inflammation, glomerular hypercellularity and ECM deposition in anti-Thy1.1 nephritic rats, MMF could be a good comparator because some studies have proven its excellent benefits on proteinuria and glomerular lesions in anti-Thy1.1 nephritic rats^{18,20}. MMF is a prodrug that can be rapidly metabolized to its active component mycophenolic acid (MPA) by the esterases in the blood⁴⁴, thus MPA was selected as the control in our study.

Reference:

- 32 Wang, W. & Chen, N. in *New Insights into Glomerulonephritis* Vol. 181 75-83 (Karger Publishers, 2013).
- 33 Koike, M. *et al.* Clinical assessment of low-dose steroid therapy for patients with IgA nephropathy: a prospective study in a single center. *Clinical and experimental nephrology* **12**, 250-255 (2008).
- 34 Pozzi, C. *et al.* Corticosteroid effectiveness in IgA nephropathy: long-term results of a randomized, controlled trial. *Journal of the American Society of Nephrology* **15**, 157-163 (2004).
- 35 Rosselli, J. L., Thacker, S. M., Karpinski, J. P. & Petkewicz, K. A. Treatment of IgA nephropathy: an update. *Annals of Pharmacotherapy* **45**, 1284-1296 (2011).
- 36 Hogg, R. J. *et al.* Clinical trial to evaluate omega-3 fatty acids and alternate day

- prednisone in patients with IgA nephropathy: report from the Southwest Pediatric Nephrology Study Group. *Clinical Journal of the American Society of Nephrology* **1**, 467-474 (2006).
- 37 Katafuchi, R. *et al.* Controlled, prospective trial of steroid treatment in IgA nephropathy: a limitation of low-dose prednisolone therapy. *American journal of kidney diseases* **41**, 972-983 (2003).
- 38 Chen, Y., Li, Y., Yang, S., Li, Y. & Liang, M. Efficacy and safety of mycophenolate mofetil treatment in IgA nephropathy: a systematic review. *BMC nephrology* **15**, 1 (2014).
- 39 Nowack, R., Birck, R. & van der Woude, F. J. Mycophenolate mofetil for systemic vasculitis and IgA nephropathy. *The Lancet* **349**, 774 (1997).
- 40 Tang, S. *et al.* Mycophenolate mofetil alleviates persistent proteinuria in IgA nephropathy. *Kidney international* **68**, 802-812 (2005).
- 41 Tang, S. C. *et al.* Long-term study of mycophenolate mofetil treatment in IgA nephropathy. *Kidney international* **77**, 543-549 (2010).
- 42 Ballardie, F. W. & Roberts, I. S. Controlled prospective trial of prednisolone and cytotoxics in progressive IgA nephropathy. *Journal of the American Society of Nephrology* **13**, 142-148 (2002).
- 43 Haubitz, M. Acute and Long-ter Toxicity of Cyclophosphamide. *Transplantationsmedizin* **19**, 26 (2007).
- 18 Suana, A. *et al.* Single application of low-dose mycophenolate mofetil-OX7-immunoliposomes ameliorates experimental mesangial proliferative glomerulonephritis. *Journal of Pharmacology and Experimental Therapeutics* **337**, 411-422 (2011).
- 20 Ziswiler, R., Steinmann-Niggli, K., Kappeler, A., Daniel, C. & Marti, H.-P. Mycophenolic acid: a new approach to the therapy of experimental mesangial proliferative glomerulonephritis. *Journal of the American Society of Nephrology* **9**, 2055-2066 (1998).
- 44 Allison, A. & Eugui, E. Purine metabolism and immunosuppressive effects of mycophenolate mofetil (MMF). *Clinical transplantation* **10**, 77-84 (1996).

b) Please cite compelling references as to the proven efficacy of TGV in glomerulonephritis or modify omit statements on page 3 lines 45-46, and lines 52-53; Page 7 lines 133-134.

Response: As we mentioned in the prior manuscript, TGV and its formulations have long been used to treat glomerulonephritis in China⁴⁵⁻⁵⁰. References were included and highlighted in red in the revised manuscript (page 3 lines 44-45).

Reference:

- 45 Chen, Y.-Z. *et al.* Meta-analysis of *Tripterygium wilfordii* Hook F in the immunosuppressive treatment of IgA nephropathy. *Intern. Med.* **49**, 2049-2055

- (2010).
- 46 Fan, W. & Wang, X. The progress in the application of tripterygium wilfordii preparations in kidney disease. *J. Clin. Nephrol.* **13**, 380-381 (2013).
- 47 Li, L. & Liu, Z. The application of Tripterygium wilfordii in the treatment of glomerulonephritis for twenty five years. *J. Nephrol. Dial. Transplant.* **12**, 246-247 (2003).
- 48 Li, L. *et al.* Clinical study of Tripterygium wilfordii Hook in treating glomerulonephritis. *Chin. J. Intern. Med.* **20**, 216-220 (1981).
- 49 Rong, S., Hu, W., Liu, Z., Tang, Z. & Li, L. A new regime of Tripterygium wilfordii Hook f. in treating primary mesangial proliferative glomerulonephritis. *J. Nephrol. Dial. Transplant.* **7**, 409-414 (1998).
- 50 Wang, C.-d. & Guo, Y.-p. The active constituents of Tripterygium wilfordii and their pharmacological actions and clinical application. *Chin. J. Intern. Med.* **1**, 235-239 (1995).

c) The significance of individual results is overstated on several occasions: for example the inference that this treatment has direct clinical relevance, that the reduction in acute proteinuria “may have profound clinical significance” in that it translates to chronic disease.

Response: Thank you very much for the thoughtful comment. We have deleted “**The toxic effects of proteinuria with abnormal filtration of proteins on the glomerular mesangium and the tubular cells, which ultimately result in glomerular sclerosis and renal failure. Therefore, the ability of CLT-AN to significantly suppress proteinuria may have profound clinical importance**” and other overstatements accordingly.

d) I am not sure that the interesting studies that they have performed on toxicity justify conclusions such as “CLT-AN represents a safe and effective treatment option for MsPGN and relevant glomerular diseases” as stated in the abstract.

Response: In our revised manuscript, we performed a comprehensive toxicity study of CLT-AN as suggested. We evaluated the systemic toxicity of CLT and CLT-AN in rats at day 1, 5, 14 or 21 post intravenous administration based on biochemical parameters and histopathological analysis (**Fig. 9; Supplementary Figs. 18-20**). CLT treatment induced severe cardiac toxicity, hepatotoxicity and neurotoxicity in rats. However, CLT-AN at the equivalent dose showed significantly reduced cardiac toxicity or hepatotoxicity, and no brain damage. Besides, given the significantly enhanced therapeutic efficacy of CLT-AN, we reasoned that the dose can be lowered to further reduce the cardiac toxicity or hepatotoxicity of CLT-AN without compromising its efficacy against glomerulonephritis. **Therefore, compared to free CLT, CLT-AN represents a promising treatment option for MsPGN and relevant glomerular diseases.** Corresponding changes were made in the **Abstract**, and highlighted in red.

7. Stylistically qualifying adjectives and adverbs that add little to the results are used: for examples “dramatic” and “dramatically” that are used on multiple occasions. Could the manuscript be modified to allow the results to stand for themselves without unnecessary commentary and qualifications.

Response: Per your suggestion, changes were made accordingly.

8. Minor comments: There are grammatical and spelling errors that should be corrected. The yellow type on a white background label in FIG 2e is not readable. Figure 10 is redundant (unless it is journal style to encourage the publication of summary cartoons).

Response: Per your suggestion,

1) We have corrected the grammatical and spelling errors carefully, and had the manuscript polished by a native speaker.

2) The yellow type on a white background label in **Fig. 2e** has been replaced with a readable **black type** in the revised manuscript.

3) **Fig. 10** was removed in our revised manuscript.

Reviewer #3 (Remarks to the Author):

This interesting study evaluates the effect of celastrol in a mesangial proliferative glomerulonephritis rat model showing that its administration in a developed mesangial cells-targeted drug delivery system via albumin nanoparticles keeps the efficacy dramatically decreasing systemic toxicity. The results regarding liver safety of the albumin nanoparticles are quite consistent; however, some aspects of liver safety assessment need needs further clarification.

1. The authors should comment the background data on hepatotoxicity of celastrol

Response: Thanks for the reviewer's comment. After a comprehensive literature survey, Huang Y *et al.* evaluated the hepatotoxicity of CLT based on histopathology examination⁵¹. Their study demonstrated that **the intraperitoneal injection** of free CLT at the dose of 1 mg/kg or 4 mg/kg induced severe lymphocyte infiltration in liver sinuses in **mice**. We have added this background data on hepatotoxicity of CLT into the **Introduction Section** in the revised manuscript (page 4 lines 59-60). In addition, we performed a toxicity study of CLT in rats **post intravenous administration based on the dosing regimen in our study**. The representative photomicrographs of main tissue sections stained with hematoxylin & eosin (H&E) and the main serum biological parameters after free CLT treatment were provided in **Fig. 2a** and **Supplementary Fig. 7**, which confirmed the hepatotoxicity of CLT, as reflected by moderate atrophy of hepatic cells and dilatation of blood sinus in LD-CLT-treated group (low dose of CLT, 1 mg/kg), and moderate edema and atrophy of hepatic cells in HD-CLT-treated group (high dose of CLT, 3 mg/kg). Next, we also evaluated the systemic toxicity of CLT and CLT-AN (**Fig. 9; Supplementary Figs. 18-20**). The serious hepatotoxicity of CLT can be significantly reduced by encapsulating free CLT into albumin nanoparticles (CLT-AN).

Reference:

51 Huang, Y. *et al.* Preparation, characterization, and assessment of the antiglioma effects of liposomal celastrol. *Anti-cancer drugs***23**, 515-524 (2012).

2. The authors selected AST and ALT as “liver function tests” (line 359), which is not correct as aminotransferases are non-specific markers of liver injury but they do not measure liver “function” (Senior ClinPharmacolTher 2012). Liver function is better assessed by bilirubin and international normalized ratio. Why did not they assess these parameters?

Response: Thanks for your constructive comment. The authors agree with the reviewer that liver function is better assessed by bilirubin and international normalized ratio. The level of serum total bilirubin (TbIL) can measure the function of the liver to clear bilirubin⁵², and international normalized ratio (INR) standardizing prothrombin time can evaluate the function of the liver to synthesize blood coagulation factors⁵³. The elevated

levels of either TBiL or INR may predict dysfunction of the liver, which is often associated with the liver injury or disease^{54,55}. Thereby, these liver function tests are usually performed for initial detection of liver injury or disease in clinic. Per your suggestion, we have supplemented a set of data of TBiL and INR in overall toxicity studies in our revised manuscript. In addition, though ALT and AST are not tests of liver function, they remain two important biochemical parameters to detect injury to hepatocytes⁵⁵. Thus, **all these four parameters including TBiL, INR, ALT and AST were used for evaluation of liver injury induced by CLT or CLT-AN (Fig. 9; Supplementary Figs. 7, 19 and 20)**. Also, histopathological analysis was also conducted to provide evidence for the results obtained from the biochemical analysis in the revised manuscript (**Figs. 2 and 9; Supplementary Figs. 18-20**).

Reference:

- 52 Thapa, B. & Walia, A. Liver function tests and their interpretation. *The Indian Journal of Pediatrics* **74**, 663-671 (2007).
- 53 Polson, J. & Lee, W. M. AASLD position paper: the management of acute liver failure. *Hepatology* **41**, 1179-1197 (2005).
- 54 Bellest, L., Eschwege, V., Poupon, R., Chazouillères, O. & Robert, A. A modified international normalized ratio as an effective way of prothrombin time standardization in hepatology. *Hepatology* **46**, 528-534 (2007).
- 55 Senior, J. Alanine aminotransferase: a clinical and regulatory tool for detecting liver injury—past, present, and future. *Clinical Pharmacology & Therapeutics* **92**, 332-339 (2012).

3. The authors shown a very good liver safety (as measured by the level of aminotransferases) of celastrol in acute studies (days 1 and 5) but chronic toxicity studies would be also required in preclinical studies.

Response: Thanks for your constructive comment. We performed the long time toxicity study of CLT or CLT-AN at day 14 or 21 post intravenous administration based on evaluation of biochemical parameters and histopathology, and added the corresponding experiments and results in the **Section “CLT-AN reduced systemic toxicity of CLT in rats”** accordingly in the revised manuscript. The main serum biological parameters and representative photomicrographs of main tissue sections stained with hematoxylin & eosin (H&E) after free CLT or CLT-AN treatment were provided in **Supplementary Figs. 19 and 20**. As they demonstrated, no obvious toxicity to spleen, lung or kidney at day 14 or 21 after CLT or CLT-AN treatment. However, severe cardiac toxicity, hepatotoxicity and neurotoxicity were observed in CLT-treated rats at day 14 or 21. CLT induced severe atrophy of myocardial cells and myofibrillar loss in the heart, diffuse edema of liver cells, red neuron and liquefactive necrosis foci in the brain. In contrast, rats treated with CLT-AN at the equivalent dose showed significantly reduced cardiac toxicity or hepatotoxicity, and no obvious signs of neurotoxicity at day 14 or 21. Overall, our study

demonstrates that CLT-AN treatment at the dose of 1 mg/kg can significantly increase the therapeutic index of CLT against anti-Thy1.1 nephritis while reducing the systemic toxicity of CLT. Given the limited time available, we will continue investigating the chronic toxicity of CLT and CLT-AN in the future study.

Reviewers' comments:

Reviewer #2 (Remarks to the Author):

The authors have provided a long and detailed response. I have the following comments (numbers relate to original review)

1) Their first author responses relate to questions about: the model and which human disease it is modelling, the meaning of proteinuria in the Thy1.1 model and the lack of an intervention study (the last addressed fully in #2)

The acute Thy1.1 model is most useful in examining the mesangial response to injury (in this case mesangiolytic, but it could in humans be by IgA deposition and its consequences), the mechanisms of mesangial cell proliferation and repair. It is correct that mesangial cell proliferation can occur in other diseases. However, the authors are incorrect in implying that mesangial cell proliferation is a major feature of diabetic nephropathy (see Najafian B et al – [including Agnes Fogo and Charles Alpers, two of the world's most eminent renal pathologists], *AJKD Atlas of Renal Pathology: Diabetic Nephropathy* AJKD 2015). The most common form of mesangial proliferative nephritis is IgA Nephropathy. The reviewer is well aware that IgA deposition is not a feature of the Thy1.1 model. But does this make the Thy1.1 an inappropriate model of IgAN? The authors seem to think so in their response, but do now mention IgAN in the revised manuscript (Discussion). They also overstate the generality and importance of mesangial proliferation in renal disease, at least as it pertains to diabetic nephropathy. While the Thy1.1 model has limitations, which can and should be acknowledged, it is a reasonable model of some renal disease. The authors should be very clear and accurate in their statements in the paper about which sort of diseases they propose that their treatment would realistically target and the caveats inherent in their studies using the Thy 1.1.

Proteinuria in the Thy1.1 model can be due to the mesangiolytic, or the proliferation and repair process that follows this. While proteinuria is important in most renal diseases, the authors fail to address the question, either in their rebuttal or in changes to the paper, of the uncertain role and meaning of proteinuria in the acute Thy1.1 model. It is noted that the authors' reason for not performing a two hit model as suggested (this would better address the role of this treatment in a progressive, not self-limited model of disease) is "limited time and financial support".

3) While the omission of primary antibody is an acceptable negative control, using an irrelevant isotype matched monoclonal (or non-immune Ig of the same species for a polyclonal antibody) is better.

4) Supplementary Table 4 is redundant as it seems the data is all presented in graphical form within the manuscript.

Reviewer #3 (Remarks to the Author):

The authors have adequately addressed the question and concerns raised by this reviewer.

Reviewer #4 (Remarks to the Author):

In this manuscript, the authors have identified celastrol, an active compound isolated from traditional Chinese medicinal plant *tripterygium wilfordii* as an efficacious agent for the treatment of mesangial proliferative glomerulonephritis. To overcome its off target toxicities, the compound was further formulated into albumin nanoparticles. The formulation showed improved efficacy and safety in an anti-Thy1.1 nephritic rat model. Further mechanistic study was also performed to understand the improved efficacy and safety. Comparing to the original version, the revision has adequately addressed the concerns of the original reviewer. One minor point the authors may want to elaborate in their discussion is the role of proximal tubule epithelium in retention of the AN in the kidney. It is not clear from the histology that the tubular region is the main accumulation

site for AN, and yet it is well recognized that the cubilin and megalin receptors expressed on proximal tubule epithelium can very effectively facilitate the recycling of albumin. There may be an additional mechanism to be considered for AN sequestration in the nephritic tissues.

Responses to the reviewers' comments:

Reviewer #2 (Remarks to the Author):

The authors have provided a long and detailed response. I have the following comments (numbers relate to original review)

1) Their first author responses relate to questions about: the model and which human disease it is modelling, the meaning of proteinuria in the Thy1.1 model and the lack of an intervention study (the last addressed fully in #2).

The acute Thy1.1 model is most useful in examining the mesangial response to injury (in this case mesangiolysis, but it could in humans be by IgA deposition and its consequences), the mechanisms of mesangial cell proliferation and repair. It is correct that mesangial cell proliferation can occur in other diseases. However, the authors are incorrect in implying that mesangial cell proliferation is a major feature of diabetic nephropathy (see Najafian B et al – [including Agnes Fogo and Charles Alpers, two of the world's most eminent renal pathologists], *AJKD Atlas of Renal Pathology: Diabetic Nephropathy* AJKD 2015). The most common form of mesangial proliferative nephritis is IgA Nephropathy. The reviewer is well aware that IgA deposition is not a feature of the Thy1.1 model. But does this make the Thy1.1 an inappropriate model of IgAN? The authors seem to think so in their response, but do now mention IgAN in the revised manuscript (Discussion). They also overstate the generality and importance of mesangial proliferation in renal disease, at least as it pertains to diabetic nephropathy. While the Thy1.1 model has limitations, which can and should be acknowledged, it is a reasonable model of some renal disease. The authors should be very clear and accurate in their statements in the paper about which sort of diseases they propose that their treatment would realistically target and the caveats inherent in their studies using the Thy 1.1. Proteinuria in the Thy1.1 model can be due to the mesangiolysis, or the proliferation and repair process that follows this. While proteinuria is important in most renal diseases, the authors fail to address the question, either in their rebuttal or in changes to the paper, of the uncertain role and meaning of proteinuria in the acute Thy1.1 model. It is noted that the authors' reason for not performing a two-hit model as suggested (this would better address the role of this treatment in a progressive, not self-limited model of disease) is "limited time and financial support".

Response: The authors would like to thank the reviewer for the thoughtful comments.

i) The authors have been well aware that mesangial cell proliferation is not a major feature of diabetic nephropathy and we deleted the statement "Mesangial hypercellularity and ECM accumulation are important features of many glomerular diseases, including IgA nephropathy, lupus nephritis and diabetic nephropathy" in the revised manuscript.

ii) Regarding the anti-Thy1.1 model and which human disease it is modelling, the authors consider it not suitable to examine the anti-IgA effects of CLT or CLT-AN based on the anti-Thy1.1 rat model because mesangial IgA deposition is not present in this model^{1,2}. The authors agree that anti-Thy1.1 model represents a reasonable model of

some renal diseases despite its limitations. Anti-Thy1.1 model can be induced in rats by either polyclonal or monoclonal antibodies against Thy1.1 antigen present on the surface of mesangial cells, in which *in situ* immune complex formation with consecutive complement-dependent mesangiolysis in the mesangium may initiate and sustain mesangial proliferation and extracellular matrix expansion³⁻⁵. Anti-Thy1.1 model is a well-characterized rat model of mesangioproliferative glomerulonephritis (MsPGN), which can simulate the immune complex-mediated MsPGN in human, such as IgA nephropathy². In our study, CLT showed potent effects against proteinuria, inflammation, glomerular hypercellularity and ECM deposition in anti-Thy1.1 nephritis through anti-inflammatory, anti-proliferative and anti-fibrotic effects, representing a promising agent for the treatment of MsPGN such as IgA nephropathy (**highlighted in the Discussion section**)

iii) Previous studies demonstrated that proteinuria is closely related with increased glomerular filtration from damages to the glomerular filtration barrier or increased intraglomerular hydraulic pressure⁶⁻⁸. In the acute Thy1.1 model, parallel to complement-dependent pathobiological cellular events including mesangiolysis, glomerular macrophage infiltration and mesangial cell proliferation⁴, the release of host materials such as complement component C5b-9, oxidants or proteases that directly damage the glomerular filtration barrier, will lead to proteinuria^{4,9}. On the other hand, glomerular injury in this model may induce proteinuria by changing the contractile function of mesangial cells, glomerular haemodynamics or tension of glomerular basement membrane^{4,7,10}. Overall, glomerular injury is the most important cause of proteinuria in anti-Thy1.1 nephritis. In our study, anti-Thy1.1 nephritic rats treated with CLT or CLT-AN demonstrate significantly reduced glomerular infiltration of circulating monocytes, glomerular hypercellularity and ECM accumulation at day 5 after disease induction, corresponding to the morphological repair of renal lesions, the increased proteinuria in these nephritic rats was also alleviated. Thus, proteinuria may be considered a sign of acute glomerular injury, and its reduction reflects the therapeutic outcome in the repair of glomerular lesions.

iv) Per your suggestion, we have performed the intervention study of free CLT or CLT-AN after a two-hit model was established in the revised manuscript (**Fig. 5; Supplementary Figs. 7-9, Supplementary Figs. 19-21**). We established the two-hit model, an irreversible model of anti-Thy1.1 nephritis by intravenously injecting anti-Thy1.1 antibody (OX7) into unilaterally nephrectomized rats, and examined the efficacy of CLT or CLT-AN treatment initiated at day 2 after disease induction on proteinuria and progressive glomerular lesions in this model. We also added corresponding experiments and results in the “**CLT attenuates glomerular lesions in anti-Thy1.1 nephritis**” and “**CLT-AN improves the efficacy of CLT in anti-Thy1.1 nephritis**” sections and **Methods section** accordingly. **For the intervention study of free CLT, CLT treatment significantly attenuated proteinuria, inflammation, glomerular hypercellularity and ECM deposition in anti-Thy1.1 nephritic rats (Supplementary Figs. 7-9).** For the intervention study of CLT-AN, compared to free CLT, CLT-AN showed better therapeutic efficacy on proteinuria and progressive glomerular lesions in

this model (**Fig. 5; Supplementary Figs. 19,20**). Together, CLT represents an effective agent in the treatment of MsPGN, and the enhanced therapeutic efficacy could be achieved by encapsulating CLT into albumin nanoparticles (ANs).

Reference:

- 1 Eitner, F., Boor, P. & Floege, J. Models of IgA nephropathy. *Drug Discovery Today: Disease Models* **7**, 21-26 (2010).
- 2 Yang, H.-C., Zuo, Y. & Fogo, A. B. Models of chronic kidney disease. *Drug Discovery Today: Disease Models* **7**, 13-19 (2010).
- 3 Bagchus, W., Jeunink, M. & Elema, J. The mesangium in anti-Thy-1 nephritis. Influx of macrophages, mesangial cell hypercellularity, and macromolecular accumulation. *The American journal of pathology* **137**, 215 (1990).
- 4 Brandt, J. *et al.* Role of the complement membrane attack complex (C5b-9) in mediating experimental mesangioproliferative glomerulonephritis. *Kidney international* **49**, 335-343 (1996).
- 5 Yamamoto, T. & Wilson, C. B. Quantitative and qualitative studies of antibody-induced mesangial cell damage in the rat. *Kidney international* **32**, 514-525 (1987).
- 6 Christensen, E. I. & Gburek, J. Protein reabsorption in renal proximal tubule-function and dysfunction in kidney pathophysiology. *Pediatric nephrology* **19**, 714-721 (2004).
- 7 Gorriz, J. L. & Martinez-Castelao, A. Proteinuria: detection and role in native renal disease progression. *Transplantation Reviews* **26**, 3-13 (2012).
- 8 Zhang, A. & Huang, S. Progress in pathogenesis of proteinuria. *International journal of nephrology* **2012** (2012).
- 9 Couser, W. New insights into mechanisms of immune glomerular injury. *Western journal of medicine* **160**, 440 (1994).
- 10 Kawachi, H. *et al.* Epitope-specific induction of mesangial lesions with proteinuria by a MoAb against mesangial cell surface antigen. *Clinical & Experimental Immunology* **88**, 399-404 (1992).

3) While the omission of primary antibody is an acceptable negative control, using an irrelevant isotype matched monoclonal (or non-immune Ig of the same species for a polyclonal antibody) is better.

Response: The authors agree with the reviewer that using an irrelevant isotype matched monoclonal antibody in the immunohistochemical staining serves a better negative control. We will take this point into account in our future studies.

4) Supplementary Table 4 is redundant as it seems the data is all presented in graphical form within the manuscript.

Response: Per your suggestion, **Supplementary Table 4** was removed in our revised manuscript.

Reviewer #4 (Remarks to the Author):

In this manuscript, the authors have identified celastrol, an active compound isolated from traditional Chinese medicinal plant *Tripterygium wilfordii* as an efficacious agent for the treatment of mesangial proliferative glomerulonephritis. To overcome its off-target toxicities, the compound was further formulated into albumin nanoparticles. The formulation showed improved efficacy and safety in an anti-Thy1.1 nephritic rat model. Further mechanistic study was also performed to understand the improved efficacy and safety. Comparing to the original version, the revision has adequately addressed the concerns of the original reviewer. One minor point the authors may want to elaborate in their discussion is the role of proximal tubule epithelium in retention of the AN in the kidney. It is not clear from the histology that the tubular region is the main accumulation site for AN, and yet it is well recognized that the cubilin and megalin receptors expressed on proximal tubule epithelium can very effectively facilitate the recycling of albumin. There may be an additional mechanism to be considered for AN sequestration in the nephritic tissues.

Response: The authors would like to thank the reviewer for the thoughtful comment. In our study, we focused our attentions on developing a mesangial cells-targeted drug delivery system via albumin nanoparticles (ANs) to improve the safety and efficacy of celastrol (CLT) against experimental mesangioproliferative glomerulonephritis (MsPGN). Compared with free CLT, CLT-loaded ANs with an average diameter of ~95 nm (CLT-AN) significantly attenuated proteinuria and glomerular lesions including inflammation, mesangial cells proliferation and ECM deposition in anti-Thy1.1 nephritic rats. We reasoned that the significantly enhanced therapeutic efficacy of CLT-AN was due to its specific and selective mesangial cellular accumulation within glomeruli, which has been confirmed by colocalized fluorescence from AN-95 and staining for the mesangial cells (Fig. 3e).

The authors agree with the reviewer that the cubilin and megalin receptors expressed on proximal tubule epithelium can effectively facilitate the recycling of albumin⁶. However, CLT-AN obtained in our study showed an average size of 95.97 ± 0.22 nm and a zeta potential of -23.2 ± 0.2 mV. The anionic charge and size limitations of glomerular filtration membrane may prevent CLT-AN to pass across the barrier into the tubular lumen and be further re-absorbed by cubilin and megalin receptors expressed on proximal tubule epithelium⁷, which has been confirmed by no fluorescent signals of AN-95 except for a weak autofluorescence in tubular region (Fig. 3c). In addition, even if in nephritis state with defects of glomerular filtration barrier, CLT-AN might be filtered into tubular lumen and re-absorbed by proximal tubular cells via cubilin and megalin mediated internalization whereby CLT-AN undergo endosomal then lysosomal degradation⁶, which may not explain the significantly enhanced therapeutic efficacy of CLT-AN against glomerular lesions in our study. Taken together, the reabsorption of CLT-AN mediated by cubilin and megalin receptors expressed on proximal tubule epithelium might not be the dominant mechanism for AN accumulation within kidney in our study.

Thus, the potential mechanisms of targeted localization of CLT-AN in mesangial

cells we proposed as follows: i) the greater blood flow rate of 400 mL/100 g per minute in the kidney than that of 100 mL/100 g per minute in the liver, together with the high hydraulic pressure in glomerular capillary, might favor CLT-AN retaining in glomeruli¹¹; ii) fenestrations of 80-150 nm of endothelium between glomerular capillaries and mesangium, might allow CLT-AN with an average diameter of ~95 nm to permeate across the endothelial fenestrations and gain access to mesangial cells¹². Meanwhile, 10 nm-size-cutoff of glomerular filtration barrier can prevent CLT-AN from filtering into Bowman's capsule space¹²; iii) these sized CLT-AN might be internalized into mesangial cells by phagocytosis¹³. A statement has been added in the **Discussion Section** (page 20 lines 419-426). For the convenience of understanding the potential process of mesangial cellular accumulation of CLT-AN, we illustrated it in Fig. 1.

Figure 1. Schematic illustration of targeted delivery of CLT by CLT-AN to mesangial cells against anti-Thy1.1 nephritis. Abbreviations: GC, glomerular capillary; BC, Bowman's capsule; P, podocyte; FP, foot processes; GBM, glomerular basement membrane; EC, endothelial cell; EF, endothelial fenestrations; GFB, glomerular filtration barrier.

Reference:

- 6 Christensen, E. I. & Gburek, J. Protein reabsorption in renal proximal tubule-function and dysfunction in kidney pathophysiology. *Pediatric nephrology* **19**, 714-721 (2004).
- 7 Gorriz, J. L. & Martinez-Castelao, A. Proteinuria: detection and role in native renal disease progression. *Transplantation Reviews* **26**, 3-13 (2012).
- 11 Shimizu, H. *et al.* siRNA-based therapy ameliorates glomerulonephritis. *Journal of the American Society of Nephrology* **21**, 622-633 (2010).
- 12 Zuckerman, J. E. & Davis, M. E. Targeting therapeutics to the glomerulus with nanoparticles. *Advances in chronic kidney disease* **20**, 500-507 (2013).

- 13 Schreiner, G. F. The mesangial phagocyte and its regulation of contractile cell biology. *Journal of the American Society of Nephrology* **2**, S74 (1992).

REVIEWERS' COMMENTS:

Reviewer #2 (Remarks to the Author):

The further experiments in the progressive model are very useful and address further the question of whether the intervention might be therapeutically useful in human disease.

Reviewer #4 (Remarks to the Author):

The authors have properly address those issues that I have raised.